

# Evaluation of lossless and lossy algorithms for the compression of scientific datasets in NetCDF-4 or HDF5 formatted files

Xavier Delaunay[1], Aurélie Courtois[1], Flavien Gouillon[2]

[1]Thales Services, 290 allée du Lac, 31670 Labège, France
[2]CNES, Centre spatial de Toulouse, 18 avenue Edouard Belin, 31401 Toulouse, France

*Correspondence to*: Xavier Delaunay (xavier.delaunay@thalesgroup.com)

**Abstract.** The increasing volume of scientific datasets imposes the use of compression to reduce the data storage or transmission costs, specifically for the oceanography or meteorological datasets generated by Earth observation mission ground segments. These data are mostly produced in NetCDF formatted files. Indeed, the NetCDF-4/HDF5 file formats are
widely spread in the global scientific community because of the nice features they offer. Particularly, the HDF5 offers the dynamically loaded filter plugin functionality allowing users to write filters, such as compression/decompression filters, to process the data before reading or writing it on the disk. In this work, we evaluate the performance of lossy and lossless compression/decompression methods through NetCDF-4 and HDF5 tools on analytical and real scientific floating-point datasets. We also introduce the Digit Rounding algorithm, a new relative error bounded data reduction method inspired by
the Bit Grooming algorithm. The Digit Rounding algorithm allows high compression ratio while preserving a given number of significant digits in the dataset. It achieves higher compression ratio than the Bit Grooming algorithm while keeping similar compression speed.

## 1 Introduction

Ground segments that process scientific mission data are facing challenges due to ever increasing resolution of on-board
instruments and data volume to: process, store and transmit. This is the case for oceanographic and meteorological missions for instance. Earth observation mission ground segments produce very large files mostly in NetCDF format: it is a standard in the oceanography field and quite spread in the meteorological community. This file format is widely spread in the global scientific community because of the nice features it offers. The fourth version of the NetCDF library, denoted NetCDF-4/HDF5 (as it is based on HDF5 layer), offers some native compression features, namely 'Deflate' and 'Shuffle' algorithms.
However, the compression performance achieved does not fully fulfil the ground processing requirements to reduce significantly the storage and dissemination cost as well as the IO times between two modules of the processing chain.

Facing the increasing volume of data, scientists are more disposed to compress data but with some requirements: science data are generally floating point data; the compression and decompression have to be fast, lossless, or lossy under some





conditions: the precision or data loss shall be controlled, the compression ratio higher than the ones of lossless algorithms. In the lossy case, there is a trade-off between the data volume and the accuracy of the compressed data.

Nevertheless, scientists can afford for small losses under the noise level in the data. Noise is indeed hardly compressible and of poor interest for the scientists, thus they do not consider as loss, data alterations that are under the noise level (Baker et al.,
5   2016).

Hence, in order to increase the compression performance within the processing chain, a degradation of the data is considered via the use of so-called "clipping" methods before the compression. Clipping methods allows increasing the compression performance by removing the least significant digits or bits in the data. Indeed, at some level, these least significant digits or bits may not be scientifically meaningful in datasets corrupted by noise, and this is particularly true for floating point data.

This paper studies compression and clipping old and new methods that can be applied to scientific datasets in order to maximize the compression performance while preserving the scientific data content and the numerical accuracy. It focuses on methods that can be applied to scientific datasets, i.e. vectors or matrices of floating point numbers.

First, lossless compression algorithms can be applied to any kind of data. The standard is the Deflate algorithm (Deutsch, 1996), native in NetCDF-4/HDF5 libraries. It is widely spread and implemented in compression tools such as zip, gzip and
zlib library. It is a reference for lossless data compression. Recently, alternatives lossless compression algorithms have emerged such as Google Snappy, LZ4 (Collet, 2013) or Zstandard (Collet and Turner, 2016). These algorithms do not make use of Huffman coding to achieve faster compression than Deflate.

Second, pre-processing methods such as the Shuffle available in HDF5 or Bitshuffle (Masui et al., 2015) allow optimizing the lossless compression by reordering the data bytes or bits in a "more compressible" order.

Third, some lossy compression algorithms such as FPZIP (Lindstrom and Isenburg, 2006), ZFP (Lindstrom, 2014) or Sz (Tao et al, 2017a), are specifically designed for the compression of scientific data, in particular floating-point data, and allow controlling the data loss.

Fourth, data reduction methods such as Linear Packing (Caron, 2014a), Layer Packing (Silver and Zender, 2017), Bit Shaving (Caron, 2014b), and Bit Grooming (Zender, 2016a) introduce some loss in the data content without necessarily
reducing the data volume. Pre-processing methods and lossless compression can then be applied to obtain higher compression ratio.

This paper focuses on compression methods implemented for NetCDF-4 or HDF5 files. Indeed, these scientific file formats are widely spread across the oceanography and meteorological community. HDF5 offers the dynamically loaded filter plugin functionality. It allows users writing filters, such as compression/decompression filters, to process the data before reading or
writing it on the disk. Consequently, many compression/decompression filters, such as Bitshuffle, Zstandard, LZ4, Sz, have been implemented by members of the HDF5 users' community and are freely accessible. On the other hand, the NetCDF Operator toolkit (NCO) (Zender, 2016b) offers some compression features such as bit shaving and Bit Grooming.

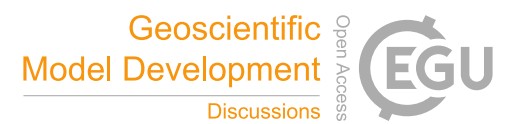

This paper is organized into five more sections. Section 2 presents the lossless and lossy compression schemes for scientific floating point datasets and the absolute and relative error bounded compression modes. Section 3 introduces the Digit Rounding algorithm. This algorithm alters the data in a relative error bounded manner to make them more compressible. It is an alternative, inspired by the Bit Grooming algorithm. Section 4 describes the performance assessment of a selection of

lossless and lossy compression methods on synthetic datasets. It presents the datasets, the performance metrics, the compression results, and finally provides some recommendations. Section 5 provides some compression results obtained on real CFOSAT and SWOT datasets. Last, section 6 provides our conclusions.

## 2 Compression algorithms

Data reduction, preprocessing and lossless coding methods can be chained as illustrated on Fig. 1.

The lossless coding step is reversible. It does not introduce any alteration in the data but allows reducing the data volume. This step can make use of lossless compression algorithms such as Deflate, Snappy, LZ4 or Zstandard.

In this paper, we choose to evaluate the performance of the lossless compression algorithms Deflate LZ4 and Zstandard; Deflate because it is the reference, LZ4 because it is a widely spread very high speed compressor and Zstandard because it is the new concurrent of Deflate, both on compression ratios and compression/decompression speeds.

Deflate make use of LZ77 dictionary coding (Ziv and Lempel, 1977) and of Huffman entropy coder (Huffman, 1952). Both methods exploit different types of redundancies. This allows Deflate achieving rather high compression ratios. However, the computational cost of the Huffman coder is high and makes Deflate compression rather slow.

LZ4 is a dictionary coding algorithm designed to provide high compression/decompression speeds rather than high compression ratio. For this, it does not make use of any entropy coder.

Zstandard is a fast lossless compressor achieving high compression ratios. It makes use of dictionary coding (repcode modelling) and of a finite state entropy coder (tANS) (Duda, 2013). It achieves similar compression ratio than Deflate with high compression/decompression speeds.

The preprocessing step is also reversible. It reorders the data bytes or bits to enhance the lossless coding step performance. It can make use of lossless compression algorithms such as Shuffle, or Bitshuffle.

The data reduction step is not reversible: data losses are introduced in this step. The strategy is to remove irrelevant data such as noise or other scientifically meaningless data. Depending on the algorithm use, this step can reduce the data volume. For instance, the Linear Packing and Sz algorithms allow reducing the data volume but not bit shaving and Bit Grooming algorithm.

One feature required for lossy the scientific data compression is the control of the amount of loss or the accuracy of the

compressed data. Depending on the data, this accuracy can be expressed by an absolute or a relative error bound.



The maximum absolute error $mae$ is defined by $mae = \max_i |\hat{s}_i - s_i|$ where the $s_i$ are the samples values of the original dataset and the $\hat{s}_i$ are the samples values of the compressed dataset. An absolute error bound specifies the maximum absolute error $e_{abs}$ allowed between any sample of the original and compressed data: $mae \leq e_{abs}$. The maximum relative error $mre$ is defined by $mre = \max_i \left| \frac{\hat{s}_i - s_i}{s_i} \right|$ A relative error bound specifies the maximum relative error $e_{rel}$ allowed between any sample

of the original and compressed data: $mre \leq e_{rel}$.

The absolute error bound can be useful for data with a unique dynamic range of interest. The relative error bound can be useful for data where both very small value and very high values are of same interest.

The Decimal Rounding algorithm (also mentioned as DSD algorithm for Decimal Significant Digit) and the Bit Grooming algorithm (also mentioned as a NSD algorithm for Number of Significant Digits) proposed in (Zender, 2016a) address both

cases. The Decimal Rounding algorithm respects a maximum error bound by preserving the specified number of decimal significant digits. The Bit Grooming algorithm respects a relative error bound by preserving the specified total number of significant digits. One interesting feature of these algorithms is the fact the accuracy of the compressed data can easily be interpreted: rather than defining the number of significant bits, they define the number of significant digit or the number of significant decimal digits.

The Bit Grooming algorithm creates a bitmask to alter the least significant bits of the mantissa of IEEE 754 floating-point data. Given a specified total number of significant digits $nsd$, the Bit Grooming algorithm tabulates the number of mantissa bits that has to be preserved to guaranty the specified precision of $nsd$ digits: to guarantee preserving 1-6 digits of precision, Bit Grooming must retain 5, 8, 11, 15, 18, and 21 mantissa bits, respectively. The advantage is that the computation of the number mantissa bits that has to be preserved is very fast. However, it is not optimal. In many cases, the number of mantissa

bits preserved is higher than what would have been strictly necessary.

Table  provides the example on the value of $\pi$ with a specified precision of $nsd = 4$ digits. The Bit Grooming algorithm preserves 15 mantissa bits where it would have been enough to preserve only 12 bits.

Optimizing the number of mantissa bits preserved will have a favorable impact on the compression ratios since it allows for zeroing more bits and thus creating longer sequences of zero bits. Thus in the next section, we propose the Digit Rounding

algorithm to overcome this limitation of the Bit Grooming algorithm.

**3 The Digit Rounding algorithm**

The Digit Rounding algorithm is similar to the Decimal Rounding algorithm in the sense that it computes a quantization factor $q$, which is a power of 2 in order to set bits to zero in the binary representation of the quantized floating point value.

The Digit Rounding algorithm makes use of a uniform scalar quantization with a reconstruction at the bins center:

$$\tilde{s}_i = \text{sign}(s_i) \times \left( \left\lfloor \frac{|s_i|}{q_i} \right\rfloor + 0.5 \right) \times q_i \tag{1}$$

where $\tilde{s}_i$ is the quantized value of the sample value $s_i$. The quantization error is bounded by:



$$|s_i - \tilde{s}_i| \leq q_i/2 \qquad (2)$$

The number of digits $d_i$ before the decimal separator in the value $s_i$ is:

$$d_i = \lfloor \log_{10}|s_i| + 1 \rfloor \qquad (3)$$

We want to preserve $nsd$ significant digits of the sample value $s$. This is approximately equivalent to having a rounding error of less than half the last tenth digit preserved. The quantization error shall thus be lower or equal to:

$$|s_i - \tilde{s}_i| \leq 0.5 \times 10^{d_i - nsd} \qquad (4)$$

Combining Eq. (2) and Eq. (4), we look for the highest quantization factor $q_i$ such that:

$$q_i/2 \leq 0.5 \times 10^{d_i - nsd}$$

or:

$$\log_{10}(q_i) \leq d_i - nsd$$

Moreover, in order to lower the computational cost and increase the compression efficiency, we look for a quantization factor that is a power of two. This allows bit-masking instead of division and creates sequences of bits 0:

$$q_i = 2^{p_i} \qquad (5)$$

We thus look for the greatest integer $p_i$ such that:

$$p_i \leq (d_i - nsd) \log_2 10$$

Finally, we take the value $p_i$ such that:

$$p_i = \lfloor (d_i - nsd) \log_2 10 \rfloor \qquad (6)$$

The log computation in Eq. (3) is the more computationally demanding. Nevertheless, optimization is possible as only the integer part of the result is useful. The optimized version implemented consists in computing the number of digits before the decimal separator $d$ from the binary exponent $e_i$ of value $s_i$: the value $s_i$ in binary representation is written:

$$s_i = \text{sign}(s_i) \times 2^{e_i} \times m_i$$

where the mantissa $m_i$ is a number between 0.5 and 1. Hence, using Eq. (3) we have:

$$d_i = \lfloor \log_{10}(2^{e_i} \times m_i) \rfloor + 1$$

$$d_i = \lfloor e_i \log_{10}(2) + \log_{10}(m_i) \rfloor + 1$$

As $-\log_{10}(2) < \log_{10}(m_i) \leq 0$, we use the following approximation in our implementation:

$$d_i \approx \lfloor (e_i - 1) \log_{10}(2) \rfloor + 1 \qquad (7)$$

It provides slightly under estimated values for $d_i$ but also a more conservative quantization allowing preserving the specified number of significant digits. This optimization slightly decreases the achievable compression ratios for strong benefits on the compression speed.

The Digit Rounding algorithm is summarized in Table 2.

Table 3 provides the result of the Digit Rounding algorithm on the value of $\pi$ with a specified precision of $nsd = 4$ digits. The Digit Rounding algorithm preserves 11 bits in the mantissa and sets the 12[th] bit to 1. Compared to the Bit Grooming





algorithm, 3 more bits have been set to 0. We have implemented the Digit Rounding algorithm as a new HDF5 dynamically loaded filter plugin to be able to apply it on datasets formatted as NetCDF-4 or HDF5 files.

## 4 Performance assessment on synthetic data

### 4.1 Performance metrics

5    A nearly exhaustive list of metrics for assessing the performance of lossy compression of scientific datasets is provided in Zchecker (Tao et al., 2017b). For the sake of conciseness, it has been chosen to present only a few of them in this paper. The following metrics have been chosen:

- the compression ratio $CR(F)$ to evaluate the size reduction as a result of the compression. It is defined by the ratio of the original file size over the compressed file size:

$$CR(F) = \frac{filesize(F_{orig})}{filesize(F_{comp})}$$

- the compression speed $CS(F)$ and decompression speed $DS(F)$ to evaluate the speed of the compression and of the decompression. They are defined by the ratio of the original file size over the compression or decompression time:

$$CS(F) = \frac{filesize(F_{orig})}{t_{comp}}$$

$$DS(F) = \frac{filesize(F_{orig})}{t_{decomp}}$$

The compression speed and the decompression speed are expressed in MB/s.

The following metrics have been chosen to assess the data degradation of the lossy compression algorithms:

- the maximum absolute error $e_{abs}^{max}$ to evaluate the maximum error between the original and compressed data. It is
15    defined as the maximum value of the pointwise absolute difference between the original and compressed data:

$$e_{abs}^{max} = \max_{i} |s_i - \tilde{s}_i|$$

- the mean error $\bar{e}$ to evaluate if any bias is introduced in the compressed data. It is defined as the mean of the pointwise difference between the original and compressed data:

$$\bar{e} = \frac{1}{N} \sum_{i=0}^{N-1} (s_i - \tilde{s}_i)$$

- the SNR to evaluate the signal to compression error ratio. It is defined by the ratio of the signal level over the root mean square compression error. It is expressed in decibel (dB):

$$SNR_{dB} = 20 \log_{10} \left( \frac{\sqrt{\frac{1}{N} \sum_{i=0}^{N-1} s_i^2}}{\sqrt{\frac{1}{N} \sum_{i=0}^{N-1} (s_i - \tilde{s}_i)^2}} \right)$$



### 4.2 Analytical datasets

Synthetic datasets with known statistics have been generated in order to test the compression algorithms under variable conditions. The following datasets have been generated:

- $s1$      a noisy sinusoid of 1 dimension,
- s3D      a noisy sinusoid pulse of 3 dimensions.

The signal $s1$ is a noisy sinusoid defined by:

$$s1(i) = c + a_1 \times \sin\left(2\pi i \frac{f_{s1}}{f_s}\right) + n(i)$$

Where $c$ is the mean value, $a_1$ is the amplitude of the sinusoid, $f_{s1}$ is its frequency and $n(i)$ is a zero mean Gaussian noise of variance 1. The signal $s1$ is generated with $c = 100$, $a_1$ computed so as to obtain a SNR of 20dB, and $\frac{f_{s1}}{f_s} = \frac{17}{19 \times 2}$. It allows having a bit more than two samples per period with a pattern reproduced every 17 periods. It is generated over $N = 2^{20}$ float sample values, each float value being encoded on 32bits. The volume of the dataset $s1$ is 4MB. The dataset and its histogram are shown in Fig. 2.

The signal $s3D$ a noisy sinusoid pulse of 3 dimensions defined by:

$$s3D(i_1, i_2, i_3) = a_2 \times \frac{\sqrt{i_1{}^2 + i_2{}^2 + i_3{}^2}}{\sqrt{L^2 + M^2 + N^2}} \times \sin\left(2\pi \sqrt{i_1{}^2 + i_2{}^2 + i_3{}^2} \frac{f_{s3D}}{f_{ech}}\right) + n(i_1, i_2, i_3)$$

Where $L, M, N$ are the 3 dimensions of the signal $s3D$, $a_2$ is the amplitude of the sinusoid, $f_{s3D}$ is its frequency and $n(i_1, i_2, i_3)$ is a zero mean Gaussian noise of variance 1

The signal $s3D$ is generated with $L = 256$, $M = 256$, $N = 2048$, $a_2$ computed to obtain a SNR of 40dB, and $\frac{f_{s3D}}{f_s} = \frac{17 \times 8}{19 \times N}$ in order to have 4 periods on the main axis. It is generated over $L \times M \times N = 2^{27}$ float sample values, each float value being encoded on 32bits. The volume of the dataset $s3D$ is 512MB. The dataset and its histogram are shown in Fig. 3.

The datasets $s1$ and $s3D$ datasets have been stored into NetCDF-4 formatted files.

### 4.3 Performance assessment of lossless compression methods

The lossless compression algorithms evaluated are Deflate and Zstandard with or without Shuffle or Bitshuffle preprocessing step. Moreover, LZ4 is evaluated but always with the Bitshuffle preprocessing step because the implementation of LZ4 we use embarks Bitshuffle.

We run lossless compression algorithm using h5repack tool from the HDF5 library in version 1.8.19, Deflate implemented in zlib 1.2.11, Zstandard in version 1.3.1 with the corresponding HDF5 filter available on the HDF web portal (http://portal.hdfgroup.org/display/support/Filters), and the implementation of LZ4 and Bitshuffle in the python package Bitshuffle-0.3.4.

The compression is performed calling h5repack tool with a command line formatted as follows:

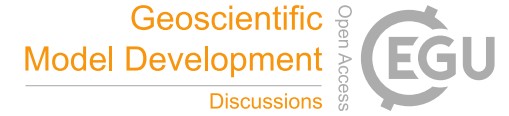



h5repack -i *in_file.nc* -o *compressed_file.h5* [--filter=*var*:params]

where *in_file.nc* is the input dataset formatted as a NetCDF-4 file and *compressed_file.h5* is the compressed dataset in HDF5 file format. The input dataset contains the *var* variable processed by one or several HDF5 filter. Table 4 provides the list of filter options used. They shall replace the filter option between brackets on previous command line.

5 The decompression is performed calling *h5repack* tool with a command line formatted as follows:

h5repack -i *compressed_file.h5* -o *out_file.h5* --filter=*var*:NONE

Compression and decompression has been performed on a Dell T1600 with an Intel Xeon E31225 4 cores CPU at 3.1GHz, and 4GB memory under RedHat 6.5 (64 bits) OS. Compression and decompression are run on a single core.

In order to obtain meaningful compression and decompression speed results, we employed the following process:

- Each compression or decompression is run 10 times.
- The elapse time (real clock time) of each run is measured.
- The minimum and maximum times measured are removed from the list of measures
- The mean of the remaining 8 measures provides the compression or decompression time.

Figure 4 provides the results obtained for the compression and decompression of the dataset *s*1 and Fig. 5 provides the 15 results obtained for the compression and decompression of the dataset *s*3*D*.

The preprocessing steps Shuffle or Bitshuffle have a favorable impact both on the compression ratio and on the compression/decompression speeds in most cases. Shuffle and Bitshuffle have similar effects on the compression performances.

The compression levels parameters *dfl_lvl* and *zstd_lvl* have little influence on the compression ratio. However, the 20 compression/decompression speeds decrease with increasing compression levels, particularly with Zstandard compression level.

The compression ratio obtained with Deflate and Zstandard are similar but the decompression speeds of Zstandard are always higher and the compression speeds of Zstandard at low compression levels are far higher.

The compression/decompression speeds obtain with Bitshuffle and LZ4 are not in all cases higher than the 25 compression/decompression speeds obtained with Bitshuffle and Zstandard at low compression level *zstd_lvl*. Nevertheless, the compression ratio obtained with Bitshuffle and LZ4 are only slightly lower than the compression ratio obtained with Bitshuffle and Zstandard at low compression level *zstd_lvl*.

Finally, the compression/decompression speeds obtained with Zstandard and LZ4 for the compression of the dataset *s*3*D* are by far lower than the one achieved for the compression of the dataset *s*1.

30 We conclude that for the lossless compression of scientific dataset the preprocessing by Shuffle of Bitshuffle are very helpful to increase the compression performance. Then, Zstandard can provide higher compression and decompression speeds than Deflate at low compression level. However, on the *s*3*D* dataset, we observe that Zstandard compression and decompression



speeds are lower than the one obtained with Deflate. Deflate and Zstandard are thus both option to consider for the lossless compression of scientific dataset but always with the Shuffle or Bitshuffle preprocessing step.

## 4.4 Performance assessment of lossy compression methods

The lossy compression algorithms evaluated are error-bounded compression algorithms. They can constrain either or both maximum absolute error or the maximum relative error.

The compression algorithms evaluated are Sz, Bit Grooming and the Digit Rounding algorithm introduced in this paper.

Sz compression algorithm has been designed to work in both error-bounded modes. Bit Grooming is declined in two algorithms: the DSD algorithm (for number of decimal significant digits) and the NSD algorithm (for number of significant digits). The DSD algorithm (also called decimal rounding algorithm) allows preserving a specific number of decimal digits. In this sense, it bounds the maximum absolute error. The NSD algorithm allows preserving a specific number of significant digits. In this sense, it bounds the maximum relative error. As the NSD algorithm, the Digit Rounding algorithm allows preserving a specific number of significant digits and bounds the maximum relative error.

Bit Grooming and Digit Rounding algorithms does not compress the data. They only alter the data to make it more compressible. Thus, lossless compression steps are required afterward. By default, Sz algorithm embark Deflate. Nevertheless, it is possible to configure Sz and deactivate Deflate to use other lossless compression algorithms.

We run Sz in version 1.4.11.1 using h5repack tool and call through its HDF5 filter plugin. We run Bit Grooming algorithms using NCO in version 4.7.0. Last, we run the Digit Rounding algorithm using h5repack tool and custom implantation of the algorithm in an HDF5 plugin filter.

Sz compression is performed calling h5repack tool with a command line formatted as follows:

h5repack -i *in_file.nc* -o *compressed_file.h5* --filter=*var*:UD=32017,0

Sz compression is configured via the *sz.config* file located in the directory from where h5repack is called. In this configuration file, quantization_intervals is set to 256 and the *szMode* is set to SZ_BEST_SPEED to achieve high speed compression. The *gzipMode* is set to Gzip_NO_COMPRESSION to deactivate Deflate compression. The *errorBoundMode* is set to ABS, or to PW_REL, to achieve respectively absolute error bounded compression, or relative error bounded compression. In the absolute error bounded compression mode, the *absErrBound* parameter is configured to achieve the desire maximum absolute error. In the relative error bounded compression mode, the parameter *pw_relBoundRatio* is configured to achieve the desire maximum relative error.

Bit Grooming compression is performed calling the *ncks* tool from NCO toolkit. The DSD algorithm is run with the following command line (note the period before the *dsd* parameter):

ncks -4 –L *dfl_lvl* --ppc *var*=**.***dsd in_file.nc compressed_file.nc*

The NSD algorithm is run with the following command line:

ncks -4 –L *dfl_lvl* --ppc *var*=*nsd in_file.nc compressed_file.nc*

In all cases, the decompression is performed calling h5repack tool with a command line formatted as follows:

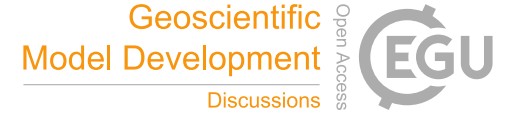



h5repack -i *compressed_file.h5* -o *out_file.h5* --filter=*var*:NONE

### 4.4.1 Performance comparison in the absolute error bounded compression mode

In order to measure the compression ratio and the compression speeds, Zstandard with *zstd_lvl* = 5 has been applied after Sz and Shuffle and Zstandard with *zstd_lvl* = 5 has been applied after Bit Grooming. This compression level provides a good

trade-off between compression speed and compression ratio.

Only Shuffle is only applied after Bit Grooming. Indeed, experiments have shown that Shuffle or Bitshuffle preprocessing do not increase the compression ratio when applied after Sz, and Bitshuffle provide lower compression ratio than Shuffle when applied after Bit Grooming.

Table 5 compares the compression performance obtained in the absolute error bounded compression mode for $e_{abs}$ = 0.5.

This correspond corresponding to *dsd* = 0 decimal significant digits preserved, or in other words, a rounding to the nearest integer.

Sz compression is performed calling h5repack tool with a command line formatted as follows:

h5repack -i *in_file.nc* -o *compressed_file.h5* --filter=*var*:UD=32017,0 --filter=*var*:UD=32015,1,5

With the *absErrBound* parameter set to 0.5 in the *sz.config* file located in the directory from where h5repack is called.

Bit Grooming compression is performed successively calling *ncks* and h5repack tool with command lines formatted as follows:

ncks -4 –L 0 --ppc *var*=*.dsd in_file.nc bitgroomed_file.nc*

h5repack -i *bitgroomed_file.nc* -o *compressed_file.h5* --filter=*var*:SHUF --filter=*var*:UD=32015,1,5

Both Sz and Bit Grooming algorithms respect the specified maximum absolute error value. Moreover, none introduces a

statistical bias: the mean absolute errors of both algorithms, not shown in this table, are very close to zero. The errors introduced by these two algorithms are similar. However, it can be shown that Bit Grooming provided higher compression ratio than Sz on the dataset $s1$, while the compression speeds are similar. On the contrary, Sz provide higher compression ratio and compression speed than Bit Grooming on the dataset $s3D$.

Figure 6 compares the performances of Sz and Bit Grooming algorithms in terms of SNR versus compression ratio. This

figure has been obtained with the following parameters:

- For Sz algorithm, the *absErrBound* parameter is successively set to 5e-5, 5e-4, 5e-3, 5e-2, 5e-1, 5
- For the Bit Grooming algorithm, the *dsd* parameter is successively set to 4, 3, 2, 1, 0, -1

As for the results reported in Table 5, Zstandard with *zstd_lvl* = 5 has been applied after Sz and Shuffle and Zstandard with *zstd_lvl* = 5 has been applied after Bit Grooming.

On the dataset $s1$, the Bit Grooming algorithm provides better compression performance than Sz except for very high compression ratio ($dsd \leq$ -1 or *absErrBound* $\geq$ 5). On the dataset $s3D$, the Bit Grooming algorithm provides better compression performance than Sz but only for low compression ratio ($dsd \geq$ 2 or *absErrBound* $\leq$ 5e-3).





We conclude that both Sz and Bit Grooming algorithms are valuable for the compression in the absolute error bounded compression mode. Bit Grooming tend to provide better performance at low compression ratios while Sz tends to provide better performance at higher compression ratios but the limit depends on the dataset.

**4.4.2 Performance comparison in the relative error bounded compression mode**

As for the performance comparison in the absolute error bounded compression mode, Zstandard with *zstd_lvl* = 5 has been applied after Sz and Shuffle and Zstandard with *zstd_lvl* = 5 has been applied after Bit Grooming in order to measure the compression ratio and the compression speeds.

We first focus on the results obtained on the dataset *s*1.

Table 6 compares the compression errors obtained in the relative error bounded compression mode. The algorithms have

been configured in order to obtain a maximum absolute error of 0.5. As the maximum absolute value in *s1* dataset is 118, the *pw_relBoundRatio* parameter in Sz is set to 0.00424 and the number of significant digits *nsd* parameter in the Bit Grooming and in the Digit Rounding algorithm is set to 3 in Table 6. However, as the Bit Grooming algorithm is too conservative, results with *nsd* = 2 are also provided.

Sz compression is performed calling h5repack tool with a command line formatted as follows:

h5repack -i *in_file.nc* -o *compressed_file.h5* --filter=*var*:UD=32017,0 --filter=*var*:UD=32015,1,5

With the *pw_relBoundRatio* parameter set to 0.00424 in the *sz.config* file located in the directory from where h5repack is called.

Bit Grooming compression is performed successively calling ncks and h5repack tool with command lines formatted as follows:

ncks -4 –L 0 --ppc *var=nsd in_file.nc bitgroomed_file.nc*

h5repack -i *bitgroomed_file.nc* -o *compressed_file.h5* --filter=*var*:SHUF --filter=*var*:UD=32015,1,5

Digit Rounding is performed calling h5repack tool with a command line formatted as follows:

h5repack -i *in_file.nc* -o *compressed_file.h5* --filter=*var*:UD=*digitRoundingID*,1,3 --filter=*var*:UD=32015,1,5

It can be observed in Table 6 that all three algorithms respect the relative error bound specified. However, as previously

mentioned the Bit Grooming algorithm is too conservative. The same is observed with the Digit Rounding algorithm for the compression of the dataset *s*1. The quality obtained with the Digit Rounding algorithm is similar to the one obtained with the Bit Grooming. Nevertheless, the compression ratio is higher.

Figure 7 (left) compares the performances of Sz, Bit Grooming and Digit Rounding algorithms in terms of SNR versus compression ratio. This figure has been obtained with the following parameters:

• For Sz algorithm, the *pw_relBoundRatio* parameter is successively set to 4.24e-6, 4.24e-5, 4.24e-4, 4.24e-3, 4.24e-2, 4.24e-1

• For the Bit Grooming algorithm, the *nsd* parameter is successively set to 6, 5, 4, 3, 2, 1

• For the Digit Rounding algorithm, the *nsd* parameter is successively set to 6, 5, 4, 3, 2, 1



As for the results reported in Table 6, Zstandard with *zstd_lvl* = 5 has been applied after Sz and Shuffle and Zstandard with *zstd_lvl* = 5 has been applied after Bit Grooming and Digit Rounding algorithms.

The Digit Rounding algorithm provides better compression performance than Sz or Bit Grooming. At high compression ratio, Sz provides similar performance as the Digit Rounding algorithm.

Figure 8 (left) compares the compression ratio obtained as a function of the parameter *nsd*, which is the user specified number of significant digit. Even if the *nsd* is not a parameter of Sz algorithm, we made the correspondence between the *pw_relBoundRatio* and the *nsd* parameters for the dataset $s1$ (*i.e.* $pw\_relBoundRatio = 4.24e^{-nsd}$) and plot the compression ratio obtained with Sz algorithm on the same figure.

It can be seen that, whatever the *nsd* specified by the user, the compression ratio obtained with the Digit Rounding are higher

than the compression ratio obtained with the Bit Grooming algorithm. It can also be seen that the compression obtained with Sz algorithm are even higher.

We now focus on the results obtained on the dataset $s3D$.

As the maximum absolute value in $s3D$ dataset is 145, the *pw_relBoundRatio* parameter in Sz is set to 0.00345 and the number of significant digits *nsd* parameter in the Bit Grooming and in the Digit Rounding algorithm is set to 3 in Table 7.

It can be observed in Table 7 that all three algorithms respect the relative error bound specified. However, on this dataset, Sz algorithm is twice too conservative. That is why, results obtained with *pw_relBoundRatio* = 0.0069 are also provided in order to obtain a maximum absolute error of 0.5. The compression ratio obtained with the Digit Rounding algorithm is higher than the one obtained with Sz.

Figure 7 (right) compares the performances of Sz, Bit Grooming and Digit Rounding algorithms in terms of SNR versus

compression ratio. This figure has been obtained with the following parameters:

- For Sz algorithm, the *pw_relBoundRatio* parameter is successively set to 6.9e-6, 6.9e-5, 6.9e-4, 6.9e-3, 6.9e-2, 6.9e-1

- For the Bit Grooming algorithm, the *nsd* parameter is successively set to 6, 5, 4, 3, 2, 1

- For the Digit Rounding algorithm, the *nsd* parameter is successively set to 6, 5, 4, 3, 2, 1

As for the results reported in Table 7, Zstandard with *zstd_lvl* = 5 has been applied after Sz and Shuffle and Zstandard with *zstd_lvl* = 5 has been applied after Bit Grooming and Digit Rounding algorithms.

For the dataset $s3D$, the Bit Grooming algorithm provides better compression performance than Sz. Nevertheless, the Digit Rounding algorithms provides compression performance very closed to the one of the Bit Grooming algorithm.

Figure 8 (right) compares the compression ratio obtained as a function of the parameter *nsd*, which is the user specified

number of significant digit. As for dataset $s1$, we made the correspondence between the *pw_relBoundRatio* and the *nsd* parameters for the dataset $s3D$ (*i.e.* $pw\_relBoundRatio = 6.9e^{-nsd}$) and plot the compression ratio obtained with Sz algorithm on the same figure.





On the dataset $s3D$, and whatever the $nsd$ specified by the user, the compression ratio obtained with the Digit Rounding algorithm are higher than the compression ratio obtained with the Bit Grooming algorithm or Sz.

We conclude that in most cases, Digit Rounding is superior to the Bit Grooming and Sz in the relative error bounded compression mode.

## 5 Application to scientific datasets

Lossless and lossy algorithms are now evaluated for the compression of scientific mission data: CFOSAT and SWOT.

### 5.1 Application to a CFOSAT dataset

The CFOSAT program is carried out through cooperation between French and Chinese Space Agencies (CNES and CNSA respectively). CFOSAT aims at characterizing the ocean surfaces to better model and predict the ocean states, and improve the knowledge in ocean/atmosphere exchanges. The CFOSAT products will help for marine and weather forecast and for climate monitoring. The CFOSAT satellite will carry two scientific payloads: SCAT, a wind scatterometer, and SWIM, a wave scatterometer to allow a joint characterization of ocean surface winds and waves. The SWIM (Surface Wave Investigation and Monitoring) instrument delivered by CNES is dedicated to the measurement of the directional wave spectrum (density spectrum of wave slopes as a function of direction and wavenumber of the waves). CFOSAT L1A product contains calibrated and geocoded waveform.

Currently, the baseline for the compression of CFOSAT L1A product involves a "clipping" method as a data reduction step, the Shuffle preprocessing and Deflate lossless coding with a compression level $dfl\_lvl$ of 3. The compression with "clipping" is liken to a compression in an absolute error bounded mode. It defines the least significant digit ($lsd$) and "clips" the data to keep only $lsd$ decimal digits. The $lsd$ is defined specifically for each variable of the dataset.

We study the performance of the following alternative compression methods:

- CFOSAT "clipping" method followed by Shuffle and Zstandard with a compression level $zstd\_lvl$ of 1 or 2 to achieve favor compression speeds;
- Bit Grooming (in the absolute error bounded compression mode) followed by Shuffle and Deflate or Zstandard.

Bit Grooming has been configured to keep the same number of decimal digits as CFOSAT "clipping" on each variable: $nsd = lsd$.

Unfortunately, Sz crashes on the compression of CFOSAT or SWOT datasets. That is why, no results with Sz are provided in the following tables.

The results for the compression of a CFOSAT L1A product of 7.34GB (uncompressed) are provided in Table 8.

Compared to the CFOSAT baseline compression, Zstandard increases the compression speed of about 40% while offering similar compression ratio. The use of Bit Grooming instead of CFOSAT "Clipping" method increases the compression ratio by a factor of 2 but decreases the compression speed by 40%. The decompression speeds are similar for all solutions. Our



recommendation is thus to use the Bit Grooming algorithm with Zstandard coding rather than the CFOSAT "Clipping" method with Deflate coding to achieve high compression ratio on this CFOSAT dataset, at the price of a lower compression speed.

## 5.2 Application to SWOT datasets

The Surface Water and Ocean Topography Mission (SWOT) is a partnership between NASA and CNES and continue the long history of altimetry missions with an innovative instrument: KaRin, a Ka band synthetic aperture radar. The launch is foreseen for 2021. SWOT addresses both oceanography and hydrology communities, measuring with a high accuracy water level of the ocean, rivers, and lakes.

SWOT has two modes of processing and thus two different types of products are generated: the high resolution products,
dedicated to hydrology and low resolution products mostly dedicated to oceanography. The Pixel Cloud product (called L2_HR_PIXC) contains data from the high-resolution (HR) mode of the KaRIn instrument. It contains information on the pixels that are detected as being over water. This product is generated where the HR mask is turned on. The Pixel Cloud product is organized into sub-orbit tiles for each swath and each pass, and this is an intermediate product between the L1 Single Look Complex products and the L2 lake/river ones. The product granularity is a tile of 64km long in along-track and
it covers either the left or the right swath (~60km wide).

The compression performance is evaluated on two different datasets:

- A simplified simulated SWOT L2_HR_PIXC pixel cloud product of 460MB (uncompressed);
- A realistic and representative SWOT L2 pixel cloud dataset in which only few attributes may be missing of 199MB (uncompressed).

The current baseline for the compression of the simplified simulated SWOT L2 pixel cloud product involves the Shuffle preprocessing and Deflate lossless coding with a compression level *dfl_lvl* of 4. However, the compression method for the official SWOT L2 pixel cloud product has not yet been defined.

We study the performance of the following compression methods:

- Shuffle and Deflate;
- Shuffle and Zstandard;
- Bit Grooming (in the absolute error bounded compression mode) followed by Shuffle and Deflate or Zstandard.
- Bit Grooming (in the relative error bounded compression mode) followed by Shuffle Zstandard;
- Bit Grooming (in the relative error bounded compression mode) followed by Shuffle Zstandard;
- Digit Rounding followed by Shuffle Zstandard.

Bit Grooming and Digit Rounding have been configured on a per variable basis to keep the precision required by the scientists on each variable.





It was not possible to evaluate the compression time needed to compress the datasets using the Digit Rounding algorithm because *h5repack* only allows defining filters parameters for a small number of variables. The way around to compute the compression ratio has been to process each variable one after the other. Nevertheless, we observed similar speeds for the compression/decompression of the largest variable of this dataset using Bit Grooming algorithm in the relative error bounded

mode or the Digit Rounding algorithm.

The results for the compression of the simplified simulated SWOT L2 pixel cloud product are provided in Table 9.

Compared to the SWOT prototype baseline compression, Zstandard increases more than 5 times the compression speed while offering similar compression ratio. The use of Bit Grooming in the absolute or relative error bounded mode, or the use of the Digit Rounding algorithm, increases the compression ratio by more than 30%, but divides by more than 3 the

compression speed. The decompression speeds are similar for all solutions. Our recommendation for the compression of this dataset is thus to use of Shuffle and Zstandard in lossless mode to achieve very high compression speed, or either the bit-grooming or the Digit Rounding algorithm to achieve slightly higher compression ratio at the price of lower compression speed.

The results for the compression of the representative SWOT L2 pixel cloud product are provided in Table 10.

Compared to Deflate, Zstandard increases more than 2.5 times the compression speed while offering similar compression ratio. The use of the Bit Grooming algorithm in the absolute error-bounded mode increases more than 2 times the compression ratio but reduces the compression speed. The compression ratios obtained in the relative error bounded mode, either with the Bit Grooming algorithm or the Digit Rounding algorithms, are not as high. The decompression speeds are similar for all solutions. Our recommendation for the compression of this dataset is thus to use the Bit Grooming algorithm

in the absolute error bounded mode to achieve high compression, at the price of a lower compression speed than the lossless solutions, considering that for SWOT product size is a driver, and considering the ration between compression time and processing time.

## 6 Conclusions

We have studied the performance of lossless and lossy compression algorithms both on synthetic datasets and on realistic

simulated datasets of future scientific satellites. The compression methods have been executed using NetCDF-4 and HDF5 tools.

It has been shown that for the lossless compression of scientific dataset the preprocessing by Shuffle of Bitshuffle is very helpful to increase the compression performance. The compression level options of Zstandard or Deflate have lower impacts on the compression ratio achieved but can significantly reduce the compression speed. Low compression levels are thus good

choices to achieve high compression speed with satisfactory compression ratio. Zstandard can provide similar higher compression speed than Deflate or LZ4 with similar compression ratios. However, on the three dimensional dataset, we have observed that Zstandard compression and decompression speeds are lower than the one obtained with Deflate. Depending on




the dataset, Deflate and Zstandard are thus both reasonable options to consider but always with Shuffle or Bitshuffle preprocessing step.

Lossy compression of scientific datasets can be achieved in two different error bounded modes: the absolute and relative error bounded mode. Sz and Bit Grooming algorithms can work in both modes. In the absolute error bounded mode, both Sz

and Bit Grooming algorithms are competitive. Bit Grooming tends to provide higher SNR than Sz at low compression ratios while Sz tends to provide higher SNR than Bit Grooming at higher compression ratios.

In the relative error bounded mode, the Digit Rounding algorithm introduced in this work provides higher compression efficiency than the Bit Grooming algorithm from which it derives. Moreover, it provides higher SNR than Sz in most cases.

Extends to this work could be to modify the implementation of the HDF5 filter for Sz to allow configuring the data loss on a

per variable basis or  to adapt the NetCDF-4 library to allow the activation of other filters, not only Shuffle and deflate.

## 7 Code and data availability

The Digit Rounding software source code and the data are currently only available upon request to Xavier Delaunay (xavier.delaunay@thalesgroup.com) or to Flavien Gouillon (Flavien.Gouillon@cnes.fr).

*Author contributions*. Xavier Delaunay designed and implemented the Digit Rounding software and wrote most of the manuscript. Aurélie Courtois performed most of the compression experiments and generated the analytical datasets. Flavien Gouillon provided the scientific datasets used in the experiments, supervised the study, contributed to its design and to the writing of the manuscript.

*Acknowledgements*. This work was funded by CNES under contract 170850/00 and realized at Thales Services. We thank Hélène Vadon, Damien Desroches and Claire Pottier for their contributions to the SWOT section and for the helpful proofreading.

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

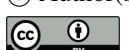



**Table 1: Representation of the value of $\pi$ in IEEE-754 single-precision binary representation (first row) and results preserving 4 significant digits with the Bit Grooming algorithm (second row) or preserving 12 mantissa bits (third row).**

| Sign | Exponent | Mantissa | Decimal | Notes |
|---|---|---|---|---|
| 0 | 10000000 | 10010010000111111011011 | 3.14159265 | Exact value of $\pi$ |
| 0 | 10000000 | 10010010000111100000000 | 3.14154053 | Result of the Bit Grooming with $nsd = 4$, 15 mantissa bits preserved |
| 0 | 10000000 | 10010010000100000000000 | 3.14111328 | Result preserving only 12 mantissa bits, allows preserving the 4 significant digits of $\pi$. |

**Table 2: The Digit Rounding algorithm.**

**Input:**

$\{s_i\}_{i=0}^n$   input sequence of samples

**Output:**

$\{\tilde{s}_i\}_{i=0}^n$   output sequence of quantized samples

**Parameter:**

$nsd$   number of significant digits preserved in each sample

**Algorithm:**

For each input sample $s_i$ in $\{s_i\}_{i=0}^n$:

1. Compute the number $d_i$ of significant digit number of digits before the decimal separator in the sample value $s_i$ following in Eq. (7)
2. Compute the quantization factor power $p_i$ following in Eq. (6)
3. Compute the quantization factor $q_i$ as in Eq. (5)
4. Compute the quantized value $\tilde{s}_i$ as in Eq. (1)

**Table 3: Representation of the value of $\pi$ in IEEE-754 single-precision binary representation (first row) and results preserving 4**
10 **significant digits with the Digit Rounding algorithm (second row).**

| Sign | Exponent | Mantissa | Decimal | Notes |
|---|---|---|---|---|
| 0 | 10000000 | 10010010000111111011011 | 3.14159265 | Exact value of $\pi$ |
| 0 | 10000000 | 10010010000100000000000 | 3.14111328 | Result of the Digit Rounding algorithm with $nsd = 4$ |



**Table 4: Command lines and parameters used for the compression with h5repack**

| Compression algorithms | Command line | Parameters |
|---|---|---|
| Deflate | --filter=*var*:GZIP=*dfl_lvl* | *dfl_lvl* from 1 to 9 |
| Shuffle + Deflate | --filter=*var*:SHUF --filter=*var*:GZIP=*dfl_lvl* | *dfl_lvl* from 1 to 9 |
| Zstandard | --filter=var:UD=32015,1,*zstd_lvl* | *zstd_lvl* from 1 to 22 |
| Shuffle + Zstandard | --filter=var:SHUF --filter=var:UD=32015,1,*zstd_lvl* | *zstd_lvl* from 1 to 22 |
| Bitshuffle + Zstandard | --filter=var:UD=32008,1,1048576 --filter=var:UD=32015,1,*zstd_lvl* | *zstd_lvl* from 1 to 22 |
| Bitshuffle + LZ4 | --filter=var:UD=32008,2,1048576,2 | - |

5 **Table 5: Compression performance of Sz and Bit Grooming in the absolute error bounded compression mode on the datasets *s1* and *s3D*.**

| | Dataset *s1* | | Dataset *s3D* | |
|---|---|---|---|---|
| Algorithm | Sz (*absErrBound* = 0.5) + Zstd (*zstd_lvl* = 5) | Bit Grooming (*dsd* = .0) + Shuffle + Zstd (*zstd_lvl* = 5) | Sz (*absErrBound* = 0.5) + Zstd (*zstd_lvl* = 5) | Bit Grooming (*dsd* = .0) + Shuffle + Zstd (*zstd_lvl* = 5) |
| Maximum absolute error | 0.5 | 0.5 | 0.5 | 0.5 |
| SNR (dB) | 30.834 | 30.830 | 45.9687 | 45.9689 |
| Compression ratio | 5.71 | 8.98 | 8.69 | 7.34 |
| Compression speed (MB/s) | 50 | 51 | 25 | 16 |

10 **Table 6: Compression performance of Sz, Bit Grooming and Digit Rounding in the relative error bounded compression mode on the dataset *s1*.**

| Algorithm | Sz | Bit Grooming | Digit Rounding |
|---|---|---|---|
| Parameter | *pw_relBoundRatio* = 0.00424 | *nsd* = 3 | *nsd* = 3 |
| Maximum absolute error | 0.5 | 0.0312 | 0.0325 |
| SNR (dB) | 30.83 | 54.93 | 54.92 |
| Compression ratio | 5.68 | 3.18 | 3.80 |

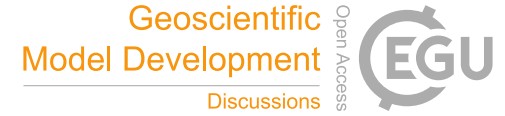



| Compression speed (MB/s) | 32 | 37 | 40 |
|---|---|---|---|

**Table 7: Compression performance of Sz, Bit Grooming and Digit Rounding in the relative error bounded compression mode on the dataset *s3D*.**

| Algorithm | Sz | | Bit Grooming | | Digit Rounding |
|---|---|---|---|---|---|
| Parameter | *pw_relBoundRatio* = 0.00345 | *pw_relBoundRatio* = 0.0069 | *nsd* = 3 | *nsd* = 2 | *nsd* = 3 |
| Maximum absolute error | 0.256 | 0.5 | 0.0625 | 0.5 | 0.5 |
| SNR (dB) | 68.06 | 62.02 | 73.96 | 55.89 | 63.94 |
| Compression ratio | 2.05 | 2.24 | 2.45 | 3.30 | 2.67 |
| Compression speed (MB/s) | 18 | 19 | 14 | 15 | 19 |

**Table 8: Performance for the compression of CFOSAT L1A products.**

| Compression method | Compression ratio | Compression speed (MB/s) | Decompression speed (MB/s) |
|---|---|---|---|
| Baseline CFOSAT compression method: CFOSAT "Clipping" + Shuffle + Deflate (3) | 5.21 | 51 (*) | 68 |
| CFOSAT "Clipping" + Shuffle + Zstandard (1) | 5.16 | 75 (*) | 67 |
| CFOSAT "Clipping" + Shuffle + Zstandard (2) | 5.38 | 72 (*) | 78 |
| Bit Grooming (abs) + Shuffle + Deflate (3) | 11.39 | 27 | 74 |
| Bit Grooming (abs) + Shuffle + Zstandard (2) | 12.68 | 35 | 81 |

(*) The time for the CFOSAT "Clipping" method is not taken into account into the compression speed computation.

**Table 9: Performance for the compression of the simplified simulated SWOT** L2_HR_PIXC **pixel cloud product.**

| Compression method | Compression ratio | Compression speed (MB/s) | Decompression speed (MB/s) |
|---|---|---|---|
| Baseline SWOT compression method: Shuffle + Deflate (4) | 14.37 | 107 | 92 |
| Shuffle + Zstandard (1) | 14.26 | 583 | 97 |
| Shuffle + Zstandard (2) | 14.36 | 589 | 97 |





| | | | |
|---|---|---|---|
| Bit Grooming (abs) + Shuffle + Deflate (4) | 20.58 | 34 | 85 |
| Bit Grooming (abs) + Shuffle + Zstandard (2) | 20.66 | 79 | 108 |
| Bit Grooming (rel) + Shuffle + Zstandard (2) | 18.87 | 50 | 101 |
| Digit Rounding + Shuffle + Zstandard (2) | 21.04 | N/A | N/A |

**Table 10: Performance for the compression of the representative SWOT L2 pixel cloud product.**

| Compression method | Compression ratio | Compression speed (MB/s) | Decompression speed (MB/s) |
|---|---|---|---|
| Shuffle + Deflate (2) | 1.98 | 52 | 83 |
| Shuffle + Zstandard (1) | 1.98 | 142 | 91 |
| Shuffle + Zstandard (2) | 1.99 | 139 | 90 |
| Bit Grooming (abs) + Shuffle + Deflate (4) | 4.42 | 40 | 98 |
| Bit Grooming (abs) + Shuffle + Zstandard (2) | 4.4 | 65 | 104 |
| Bit Grooming (rel) + Shuffle + Zstandard (2) | 2.56 | 43 | 93 |
| Digit Rounding + Shuffle + Zstandard (2) | 2.85 (*) | N/A (*) | N/A (*) |



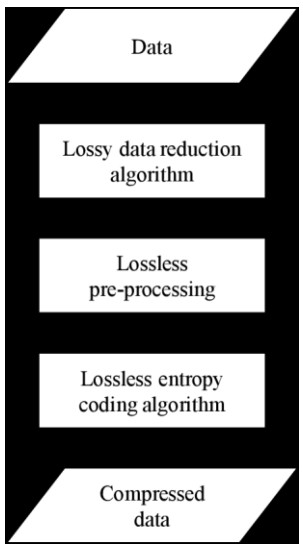

**Figure 1: Compression chain in which appears data reduction, pre-processing and lossless coding steps.**

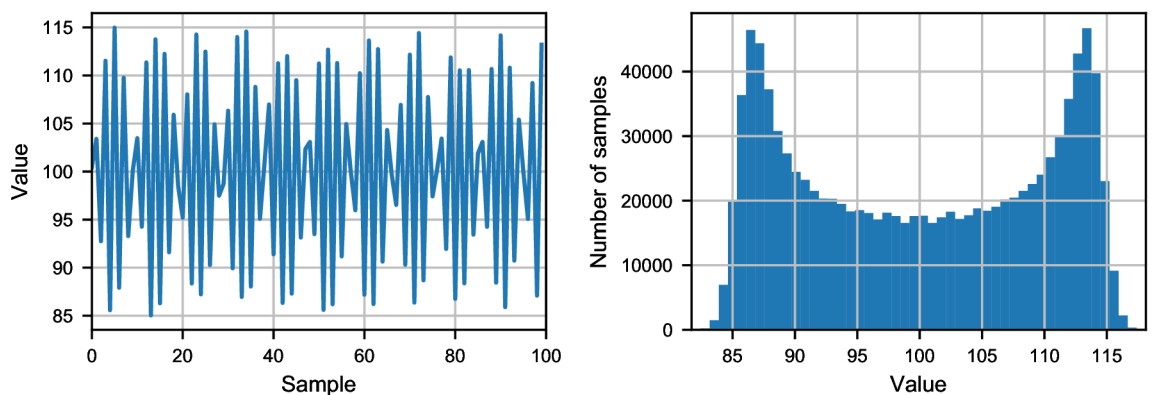

**Figure 2: First 100 samples of the dataset s_1 (left) and histogram of the sample values (right).**





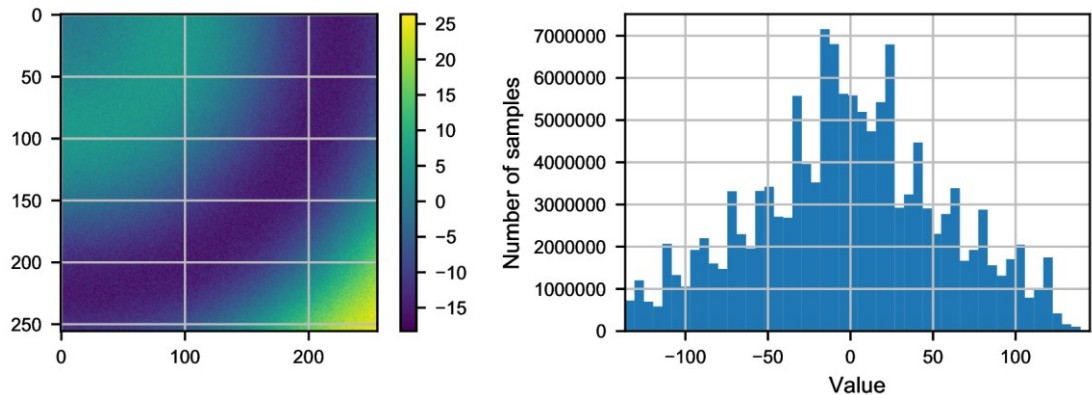

**Figure 3: Representation of the first slices $s3D(i_1, i_2, 0)$ (left), and histogram of the sample values (right).**

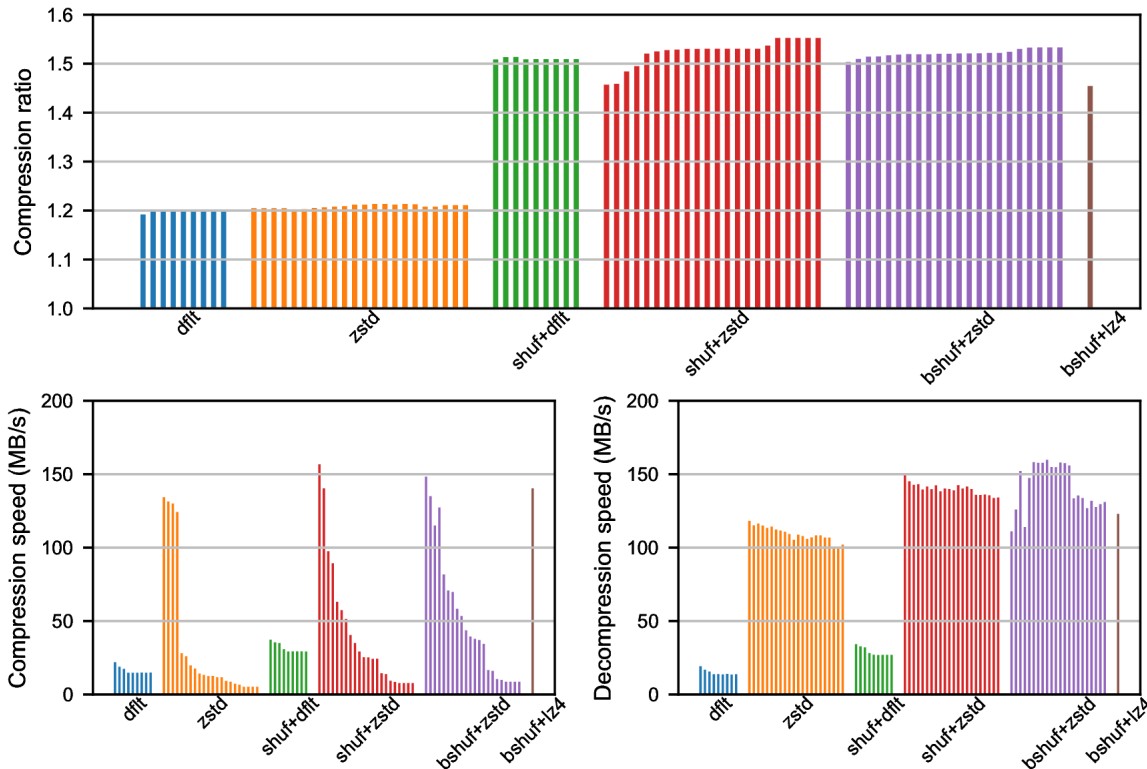

**Figure 4: Performance obtained for the lossless compression of $s1$ dataset with Deflate (dflt), Zstandard (zstd), Shuffle and Deflate (shuf+dflt), Shuffle and Zstandard (shuf+zstd), Bitshuffle and Zstandard (bshuf+zstd), Bitshuffle LZ4 (bshuf+lz4). Compression ratios (top), Compression speeds (bottom-left), decompression speeds (bottom-right).**



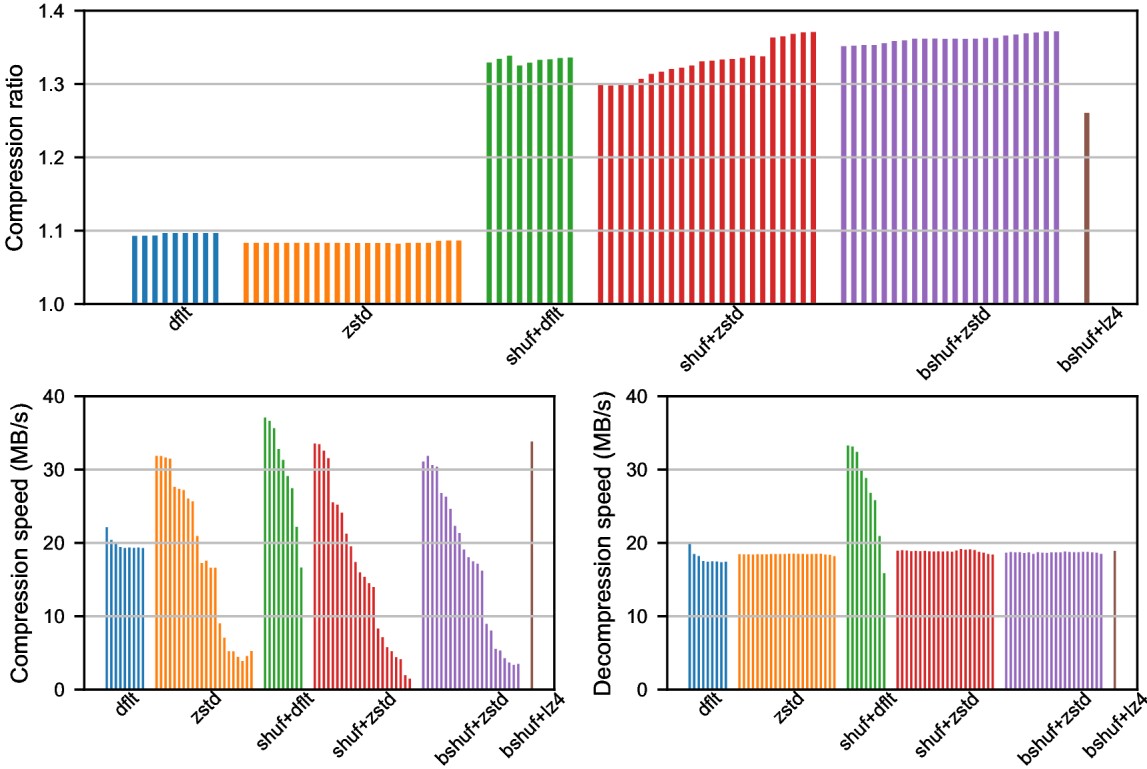

**Figure 5: Performance obtained for the lossless compression of *s3D* dataset with Deflate (dflt), Zstandard (zstd), Shuffle and Deflate (shuf+dflt), Shuffle and Zstandard (shuf+zstd), Bitshuffle and Zstandard (bshuf+zstd), Bitshuffle LZ4 (bshuf+lz4). Compression ratios (top), Compression speeds (bottom-left), decompression speeds (bottom-right).**

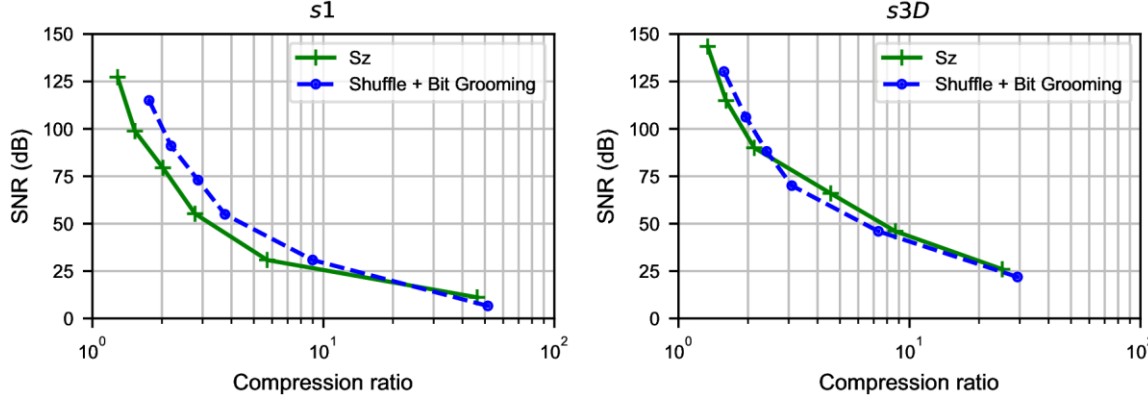

**Figure 6: Comparison of the compression performance (SNR vs. compression ratio) of Sz and Bit Grooming algorithms in the absolute error-bounded compression mode. Compression performance obtained on *s1* dataset (left) and *s3D* dataset (right).**



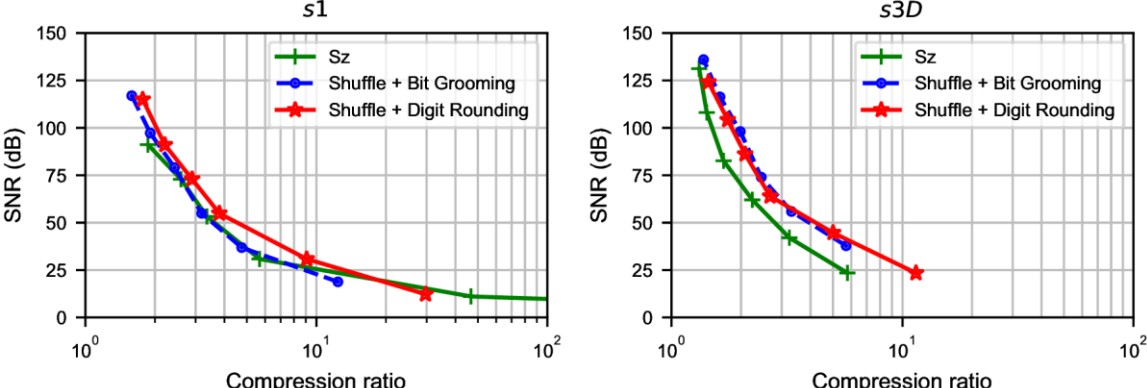

**Figure 7: Comparison of the compression performance (SNR vs. compression ratio) of Sz, Bit Grooming and Digit Rounding algorithms in the relative error-bounded compression mode. Compression performance obtained on *s1* dataset (left) and *s3D* dataset (right).**

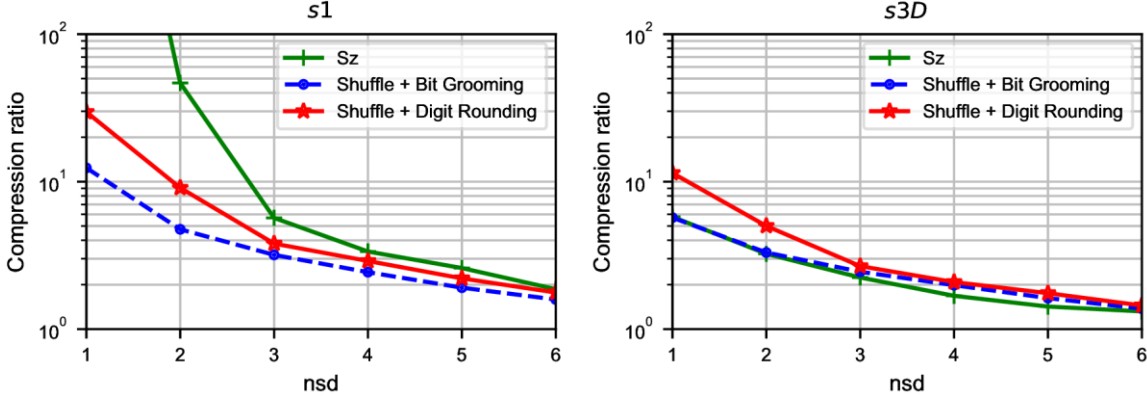

**Figure 8: Compression ratio as a function of the user specified number of significant digit (*nsd*) for Sz, Bit Grooming and Digit Rounding algorithm. Compression performance obtained on *s1* dataset (left) and *s3D* dataset (right).**