# Peer review of "Evaluation of lossless and lossy algorithms for the compression of scientific datasets in netCDF-4 or HDF5 files"

_Geoscientific Model Development, 2018_

## Referee Comment (RC1) · Anonymous Referee #1 · 16 Jan 2019

Manuscript: "Evaluation of lossless and lossy algorithms for the compression of scientific datasets in NetCDF-4 or HDF5 formatted files"

Authors: Xavier Delaunay, Aurélie Courtois, and Flavien Gouillon

——————————- Summary ——————————-

The authors focus on data compression (lossless and lossy) of HDF5 or NetCDF4 files. They compare several techniques on synthetic data and mission data and also suggest an alternative lossy method called Digit Rounding (to improve upon Bit Grooming). Data compression of scientific data is an important topic that is continuing to receive attention. This manuscript investigates compression on some synthetic data and

application data, and draws general conclusions based on those studies.

————————————- General comments ——————————

- This manuscript needs a lot of improvement in terms of the grammar and writing. There are many awkward phrases and incorrect word choices that need to be improved (a subset are listed below). The paragraph structures are also in need of modification (many paragraphs contain only 1 or 2 sentences).

-Section 2: I'd be helpful to include more detail for the preprocessing algorithms: shuffle and bitshuffle. Also this section in general needs improvement. It's a bit "choppy" to read (needs smoother and better transitions between topics) and feels like more details would be helpful on the methods (especially the ones that the digit rounding algorithm builds on).

-Section 4: Why does using the synthetic data in 4.1 to assess performance make sense - it seems unrelated to the application area of interest. I'd argue that the metrics used in 4.1 are really minimal requirements as well. Also take care when referring to "performance" as it is overloaded term...do you mean speed or effectiveness (it's used both ways)

-fpzip is a fast and effective lossless method that would have been nice to compare (I *think* there is an fpzip filter available). Also I believe that any hdf5 filter can be accessed through NetCDF4 (see last sentence in conclusion) - consider contacting the Unidata folks.

-Comments on doing compression in parallel?

-When reading the conclusion, it's hard to see what the main contributions of this paper are. It's fairly well known already that preprocessing of scientific data (e.g., bit shuffle or shuffle) improves lossy compression. Also the statements in the conclusion aren't specific to a particular type of data set, but are presented as more general conclusions. Given that the effectiveness and performance (speed) of lossy and lossless compression are very data, application, and variable dependent, the general statements here are not well justified by the small sample of data in the paper. I'd suggest focusing the paper more heavily on the data in Section 5 (if it's of interest) and tailoring the discussion in that manner. Or maybe the focus was to be more on speeds than quality, in which case it's be important to work to get sz and fpzip working, particularly via netcdf-4...

————————————— Specific items: —————————————

-p2, line 20: note that fpzip can also be lossless

-p. 8: discussion of figure 5: is the width of the bars related to the compression levels? (e.g. line 20 statement is unclear)

-p.8, lines 28-29: Why is this the case? (Add some discussion beyond describing the figure.)

-p.8, line 20: I feel like the parameters should be better explained for –filter so that the reader can try them more easily. For example, what does the "32017" mean? I think that the following 0 is for sz, but this is not stated either.

-p. 10, line 6: re: "experiments have shown" - whose or which experiments (cite?)

-Table 5 : Why is the speed faster for 1d?

-Section 4.4.1, line 21-24: Any idea why you get these results?

-Section 4.4.1, last sentences: It's unclear to me what the value of these synthetic data sets is - especially given the statement on p. 11, line 3, about the dependence on the dataset

-page 11, line 9-10: I'd include characteristics of the data (e.g., maximum abs. value) earlier in the text when the two datasets are introduced.

-page 11, line 24: I don't see relative error mentioned in Table 6 - it seems to just be

absolute error

-p.11-12: Need more of a discussion of the results in Figure 7. For 3D, it looks like bit grooming and digit rounding are similar - I don't see a clear advantage.

-p.12, lines 16-17: SZ compression can be controlled with an absolute error bound, so why is the relative error bound adjusted to get the desired abs. error?

-Section 5.1: It is disappointing not to have SZ results on the real data of interest. Were the SZ authors contacted? I would think that they could have helped resolve this issue.

-p. 15, line 18: "which only a few attributes may be missing" - It's unclear what this means. It's super helpful to really detail the data being compressed so that one can make sense of the results.

-p. 14, line 30: Please share more specific information about the precision required by the scientists for the data. Again, more information is useful for interpreting results.

-Section 5 seems like it should be the highlight of the paper as here we are seeing the results on the real data. But it feels like more detail is needed on the data and more discussion of the implications of the results.

———————————— Typos, etc.: ————————————

-abstract, line 7: incorrect use of "imposes"

-p.1, line 22: "quite spread"=> "quite prevalent" or "quite popular" , "widely spread" => "widely used"

-p.1., line 26: "reduce significantly" => "significantly reduce"

-p.1. line 27: This sentence (that continues to page 2) is too long.

-p.2, line 3: "can afford for" is awkward

-p2, lines 10-24: this region is 5 paragraphs

[Figure]

-p.3, lines 3-4: awkwardly worded

-p. 3, line 9: one sentence paragraph

-p.3, line 12: missing "," after "Deflate"

-p.3 line 14: not sure what is meant by "new concurrent"

-p.3, line 13: "widely spread" => "widely used"

-p.3, line 16: awkward sentence: "This allows Deflate achieving rather high compression ratios"

-P.3, section 3: again, there are too many tiny paragraphs

-p.4, line 7: "are of same interest" is awkward

-p.4, line 21: Table number is not given

-p.5, line 15: awkwardly worded

-p.5, line 24: One sentence paragraph

-p.7, section 4.2: define f_s, f_ech

-p.7, line 22: "use embarks" is awkward

-p.8, line 7: "declined" doesn't make sense

-p.8, line 14: "embark" - incorrect usage

-p.10, line 10: "This correspond corresponding" needs to be fixed

-p. 10, line 24: Note sure I'd use "performances" here as earlier it was used to indicate speed.

-p. 15, line 21: "ration" => "ratio"

-p.15, line 14: another one sentence paragraph
-p.16, line 9: "Extends to this work" - awkwardly worded

---

## Referee Comment (RC2) · Zender (Referee) · 26 Jan 2019

I have voluntarily disclosed my identity in all manuscript reviews since 2004. The authors are free to contact me at zender@uci.edu.

**General Comments**

This manuscript presents a new lossy compression algorithm called "Digit Rounding" (DR), and evaluates its performance against and with other lossy and lossless compression algorithms on idealized and remote sensing datasets. The manuscript addresses the growing need to archive meaningful data rather than noise, and to do

so reliably and quickly. The study presents an original advance in lossy compression whose implementation unfortunately hampers its utility. The study is understandable yet poorly written. This potentially useful study of lossy compression techniques needs a thorough overhaul before publication.

**Specific Comments**

Originality: DR is an improvement on "Bit Grooming" (BG) which I invented as an improvement on "Bit Shaving". In that sense I am qualified to comment on its originality. The heart of DR is essentially a continuous version of BG: Whereas BG fixes the number of bits masked for each specified precision, and masks these bits for every value, DR recomputes the number of bits masked for each quantized value to achieve the same precision. BG did not implement the continuous method because I thought that computing the logarithm of each value would be expensive, inelegant, and yield only marginally more compression. However, DR cleverly uses the exponent field instead of computing logarithms, and so deciphers the correct number of bits to mask while avoiding expensive floating point math. This results in significantly more compressibility that (apparently) incurs no significant speed penalty (possibly because it compresses better and thus the lossless step is faster?). Hence DR appears to be a significant algorithmic advance and I congratulate the authors for their insight.

The manuscript stumbles in places due to low quality English, and cries out for more fluent editing. Not only is the word choice often awkward, but the manuscript is like a continuously choppy sea of standalone sentences with few well developed paragraphs that swell with meaning then yield gently to the next idea. GMD readers deserve and expect better.

Does DR guarantee that it will never create a relative error greater than half the value of the least significant digit? BG chooses the number of digits to mask conservatively, so it can and does guarantee that it *always* preserves the specified precision. Equations

(1)-(7) imply that DR can make the same claim, but this claim is never explicitly tested or made. The absence of this guarantee is puzzling because it would strengthen the confidence of users in the algorithm. However, the guarantee must be explicitly tested, because it undergirds the premise that the comparison between DR and BG is fair. In any case, clearly state whether DR ever violates the desired precision, even if that happens only rarely.

p. 16 L13: "Code and data availability: The Digit Rounding software source code and the data are currently only available upon request to Xavier Delaunay (xavier.delaunay@thalesgroup.com) or to Flavien Gouillon (Flavien.Gouillon@cnes.fr)." The GMD policy on code and data is here: https://www.geoscientific-model-development.net/about/code_and_data_policy.html. This manuscript provides no code access nor explanation, and no dataset access, and thus appears to violate GMD policy in these areas.

Common comparisons would help build confidence in your results. It would have been more synergistic to evaluate the algorithms on at least one of the same datasets as Zender (2016), which are all publicly available. I am glad the authors used the publicly available NCO executables. Why not release the DR software in the same spirit so that the geoscience community can use (and possibly improve) it?

The lossless and lossy compression algorithms analyzed seem like a fairly balanced collection of those most relevant to GMD readers. Most methods that were omitted are, to my knowledge, either non-competitive (e.g., Packing) or not user-friendly, e.g., research grade but not widely available (e.g., Layer Packing) and too hard to independently implement.

Table 6 on p. 19 shows the maximum absolute error (MAE) of BG is quite similar to DR, as I would expect. However, Table 7 on p. 20 shows the maximum absolute error (MAE) of BG is nearly 10x less than DR. Why are the MAEs similar for dataset s1

and significantly different for dataset s3D? I expect DR has a greater mean error (and lower SNR) than BG due to the algorithms, yet the difference in MAEs surprises me. Zender (2016) Table 3 shows that BG is tuned to have an MAE just shy of violating the precision guarantee. An MAE that is nearly 10x larger seems like it might violate the precision guarantee.

The preceding comment is a request to more carefully analyze the underlying cause of the behaviors reported in the data. The next two comments are to report more results to deepen the analyses and explain the behavior of DR more robustly.

Please include the maximum absolute error or maximum absolute relative error (which normalizes the error by the original value) to Tables 5–10.

MeanAE is an important statistic that is complementary to MaxAE. MeanAE is the average absolute (no compensation between positive and negative) bias in the dataset, and is more familiar and relevant than SNR to at least some geophysicists. Please consider including MeanAE in Tables 5–10.

Zender (2016) and Silver and Zender (2017) consider four primary criteria to evaluate compression algorithms: Compression Ratio, Accuracy, Speed, and User-friendliness. This manuscript neglects explicit consideration of the last, though usability seems (in addition to performance) seems to be an implicit reason why they recommend BG not DR for the "real world" use cases in Sections 5.1 and 5.2. The manuscript would benefit from a more explicit consideration of usability throughout. Examples include software availability, flexibility, and complexity of invocation, as well as transparency (will users have all the necessary software required to read the compressed data?), and instructions to mitigate these issues for DR.

Tables 1 and 3 follow Tables 1 and 2 of Zender (2016). This should be noted in the text and/or caption of the tables.

It seems like Table 2, the algorithm description, should be a figure rather than a table.

The manuscript is awkward in that it introduces a demonstrably superior lossy compression algorithm but recommends a different algorithm (BG) for "real world" cases (Section 5), partly because DR is unavailable in software that potential users have easy access to, and its implementation appears to be too inflexible to use on generic datasets. The recommendation of BG not DR does attest to the objectivity of the study, yet it seems to be an unsatisfying conclusion to what was clearly a time-consuming study. In this sense the manuscript seems premature, since if DR were "ready for primetime" then the authors could have recommended it rather than BG in Section 5. Perhaps the authors should re-evaluate whether the manuscript is premature, i.e., whether it should both introduce a new lossy algorithm before it is ready to use in optimized workflows for generic geoscientific data compresssion.

**Minor Suggestions**

p. 1 L22: "well spread"

p. 2 L22: DEFLATE

p. 4 L1: $\max_i$ is redundant. Just use $\max$.

p. 4 L21: Table 1

p. 9 L7: "declined"?

p. 9 L14: "By default, Sz algorithm embark Deflate." is awkward.

p. 14 L27–28: These lines are identical

p. 18 L8: "the number $d_i$ of significant digit number of digits"???

p. 18 L8: "following Eq." not "following in Eq."

p. 23 Figure 4: Clarify the meaning of the distinct vertical bars.

---

## Author Response (AR1)

Dear Editor,

Thank you so much for extending our time for the revision of the manuscript. We also like to thank both reviewers for their thorough review of the manuscript. Below, find the main modifications made to the manuscript which covers all the remarks.

- As suggested by the reviewers, the manuscript has been reworked to focus more on the application to CFOSAT and SWOT datasets. We have also moved the details concerning the synthetic datasets in the Supplement which is now provided with the paper.

- The term "Bit Grooming" was employed both for the absolute and the relative error bounded compression modes. But the algorithms are different. In the revised version of the paper we employ the term "Decimal Rounding" in the absolute error bounded mode and the term "Bit Grooming" in the relative absolute error bounded mode to remove any ambiguities.

- Thanks to a remark from C. Zender, we found that the Digit Rounding algorithm was sometimes too conservative. We thus slightly modified the implementation. The compression results reported in the new version of the paper have been obtained with the new version of the Digit Rounding algorithm.

- In the previous version of the paper, we cascaded a call to ncks with a call to h5repack to perform Bit Grooming followed by Zstandard compression. For fairer comparisons on the compression speed, we modified our approach and now only employ ncks tool to run Decimal Rounding and Bit Grooming. However, we could not call Zstandard compression via the ncks tool, but only Deflate compression. Consequently, we now provide results for Decimal Rounding, Bit Grooming, Sz and Digit Rounding followed by Deflate compression.

- The Digit Rounding software source code is now available from CNES GitHub at https://github.com/CNES/Digit_Rounding.

- We now provide a Supplement which details the datasets and provides the command lines used for running the compression tools.

- The grammar and English of the paper has been dramatically improved by reviews from native English speakers.

**Reply to Anonymous Referee #1**

We are grateful to the referee for her/his constructive and thorough criticism and suggestions to our manuscript. Please, find below a detailed point-by-point reply. *Referee's comments are in blue italic*; our answers are in black and our changes to the manuscript in green.

*- This manuscript needs a lot of improvement in terms of the grammar and writing. There are many awkward phrases and incorrect word choices that need to be improved (a subset are listed below). The paragraph structures are also in need of modification (many paragraphs contain only 1 or 2 sentences).*

We will improve grammar and writing of the manuscript by contacting a native English speaker/writer. Thank you for pointing out the subset of incorrect word choices. We will also modify the paragraph structures to avoid too small paragraphs.

The grammar and writing have been improved throughout the manuscript. Short paragraphs have also been modified.

*-Section 2: I'd be helpful to include more detail for the preprocessing algorithms: shuffle and bitshuffle. Also this section in general needs improvement. It's a bit "choppy" to read (needs smoother and better transitions between topics) and feels like more details would be helpful on the methods (especially the ones that the digit rounding algorithm builds on).*

More details will be added on the shuffle and bitshuffle algorithms. More details will also be added on the bit-grooming and decimal rounding algorithm, algorithm on which the digit rounding algorithm is built. We will do what is needed to improve this section in general.

Section 2 has been restructured so as to make the reading smoother. Details have been added on the shuffle and bitshuffle algorithms as well as on the bit-grooming and decimal rounding algorithm.

"The Shuffle algorithm groups all the data samples' first bytes together, all the second bytes together, etc. In smooth datasets, or datasets with highly correlated consecutive sample values, this rearrangement creates long runs of similar bytes, improving the dataset's compression. Bitshuffle extends the concept of Shuffle to the bit level by grouping together all the data samples' first bits, second bits, etc.

…

The Decimal Rounding algorithm achieves a uniform scalar quantization of the data. The quantization step is a power of 2 pre-computed so as to preserve a specific number of decimal digits. The Bit Grooming algorithm creates a bitmask to degrade the least significant bits of the mantissa of IEEE 754 floating-point data. Given a specified total number of significant digits, $nsd$, the Bit Grooming algorithm tabulates the number of mantissa bits that has to be preserved to guarantee the specified precision of $nsd$ digits: to guarantee 1-6 digits of precision, Bit Grooming must retain 5, 8, 11, 15, 18, and 21 mantissa bits respectively. The advantage is that the number of mantissa bits that must be preserved is computed very quickly. The disadvantage is that this computation is not optimal. In many cases, more mantissa bits are preserved than strictly necessary."

*-Section 4: Why does using the synthetic data in 4.1 to assess performance make sense - it seems unrelated to the application area of interest. I'd argue that the metrics used in 4.1 are really minimal requirements as well. Also take care when referring to "performance" as it is overloaded term...do you mean speed or effectiveness (it's used both ways)*

The objective of using synthetic data was to control the data parameters, such as the SNR, to be able to assess the impact of these parameters on the compression ratios. The results are not reported in this paper which rather focuses on providing a comparison of the compression ratio and speed of different algorithms. It has also been chosen to present only the minimal set of relevant metrics to avoid overloading the paper. We will be more rigorous and replace the term "performance" by "compression ratio" or "compression speed" in the text.

All the occurrences of the term "performance" have been checked and corrected when needed. The Mean Absolute Error metric has been added to the list of metrics:

"mean absolute error $\overline{e_{abs}}$ to evaluate the mean data degradation. It is defined as the mean of the pointwise absolute difference between the original and compressed data:

$$\overline{e_{abs}} = \frac{1}{N} \sum_{i=0}^{N-1} |s_i - \tilde{s}_i|"$$

*-fpzip is a fast and effective lossless method that would have been nice to compare (I \*think\* there is an fpzip filter available). Also I believe that any hdf5 filter can be accessed through NetCDF4 (see last sentence in conclusion) - consider contacting the Unidata folks.*

Thank you for the suggestion. Indeed a HDF5 filter is accessible for fpzip. However, many lossless compression algorithms exist. In our paper, we chose to evaluate the most "popular", i.e. the lossless compression algorithms the most used in applications. Thank you for pointing this evolution of the NetCDF-4 library: from version 4.6.0 - January 24, 2018, NetCDF fully supports HDF5 dynamic filters. The text of the paper will be modified so as to provide the example usage using the new NetCDF-4 features.

An example using the new NetCDF-4 features with *nccopy* tool has been provided in the supplement and the conclusion has been modified to remove the last sentence. However, nccopy tool does not allow yet linking together different filters.

*-Comments on doing compression in parallel?*

We do not consider running compression algorithm in parallel in this work and will make it clear in the manuscript. It is a possible extension of this study.

The following sentence has been added in section 4:

"Parallel compression has not been considered in this work."

*-When reading the conclusion, it's hard to see what the main contributions of this paper are. It's fairly well known already that preprocessing of scientific data (e.g., bit shuffle or shuffle) improves lossy compression. Also the statements in the conclusion aren't specific to a particular type of data set, but are presented as more general conclusions.*

*Given that the effectiveness and performance (speed) of lossy and lossless compression are very data, application, and variable dependent, the general statements here are not well justified by the small sample of data in the paper. I'd suggest focusing the paper more heavily on the data in Section 5 (if it's of interest) and tailoring the discussion in that manner. Or maybe the focus was to be more on speeds than quality, in which case it's be important to work to get sz and fpzip working, particularly via netcdf-4...*

Thank you for the suggestion that will help highlighting the main contributions of our work. As suggested, the paper will be reworked to focus more on the application to the CFOSAT and SWOT datasets. We will also avoid general statements but attach our conclusions to our application case.

The paper has been reworked to focus more on the application to the CFOSAT and SWOT datasets. Sz compression has been run on CFOSAT dataset and on some parts of SWOT datasets. The NetCDF-4 tool nccopy does not allow yet linking together different filters. This restrains its usability and this is why we prefer using h5repack tool.

————————————- Specific items: ————————————-

-p2, line 20: note that fpzip can also be lossless

Thank you for the remark. Fpzip will be presented both as a lossless and lossy compression algorithm.

The text has been modified as follows:

"Third, some lossy/lossless compression algorithms, such as FPZIP (Lindstrom and Isenburg, 2006), …"

-p. 8: discussion of figure 5: is the width of the bars related to the compression levels? (e.g. line 20 statement is unclear)

No. All the bars have the same width. Each vertical bar represents a compression level. For instance, the 9 compression levels of Deflate are represented by 9 vertical bars. This will be clarified in the text p.8.

Clarifications have been added to the text:

"The vertical bars represent the results for different compression levels: from 1 to 9 for the Deflate level dfl_lvl, from 1 to 22 for Zstandard level zstd_lvl, and only one level for LZ4."

-p.8, lines 28-29: Why is this the case? (Add some discussion beyond describing the figure.)

These lower compression/decompression speeds are not well understood and would require further investigation to be fully understood. It might be related to HDF5 chunking. Indeed, HDF5 split the data into chunks of small size that are independently compressed. This allows HDF5 to improve partial I/O for big datasets but can sometimes reduce the compression/decompression speeds. This discussion will be added to the text.

The following sentence has been added to the text:

"Further investigations are required to understand why the compression/decompression speeds are lower, but it might be related to HDF5 chunking."

*-p.8, line 20: I feel like the parameters should be better explained for –filter so that the reader can try them more easily. For example, what does the "32017" mean? I think that the following 0 is for sz, but this is not stated either.*

We will add the meaning of each parameters. Each HDF5 filter is identified by a unique ID. "32017" is the identifier of Sz filter. The following "0" is the number of filter parameters. In the case of Sz, the filter does not have any parameter to set. That is why there are 0 parameters. Sz compressor is configured via the sz.config file. The same explanations will be added for the other filters used in the paper.

All the command lines have been moved to the Supplement with the explanation above to avoid overloading the manuscript.

*-p. 10, line 6: re: "experiments have shown" - whose or which experiments (cite?)*

It is based on our own experiments that haven't been published. The sentence will be reworked as follows: "We have found that Shuffle or Bitshuffle preprocessing do not increase the compression ratio when applied after Sz. We have also found that and Bitshuffle provide lower compression ratio than Shuffle when applied after Bit Grooming. That is why only Shuffle is applied after Bit Grooming."

This sentence has been removed since the results have not been published.

*-Table 5 : Why is the speed faster for 1d?*

As previously, the lower compression/decompression speeds obtained with the dataset s3D are not well understood and might be related to HDF5 chunking. This discussion will be added to the text.

The following sentence has been added to the text:

"The lower compression/decompression speeds obtained with Sz on the dataset s3D are not well understood and might be related to HDF5 chunking as previously mentioned."

*-Section 4.4.1, line 21-24: Any idea why you get these results?*

Sz performs better on smooth signals since it makes use of a prediction step. The signal s1 being highly noisy, Sz prediction might often fail. This can explain the lower compression ratio on the signal s1. On the contrary, Bit-grooming does not makes any prediction. This can explain why it achieves better compression than Sz on the signal s1. This hypothesis will be added to the text.

The following sentences have been added to the text:

"Sz may perform better on dataset $s3D$ because it is smoother than dataset $s1$. Indeed, Sz integrates a prediction step. This prediction might often fail because dataset $s1$ is very noisy. This may explain the lower compression ratio for this dataset. Decimal Rounding, however, does not make any predictions, which may explain why it achieves a better compression than Sz for dataset $s1$."

*-Section 4.4.1, last sentences: It's unclear to me what the value of these synthetic data sets is - especially given the statement on p. 11, line 3, about the dependence on the dataset*

As suggested previously, the paper will be reworked to focus more on the application to CFOSAT and SWOT datasets without drawing general conclusions based on the results obtained on the synthetic datasets.

We modified the last sentence of this section as follows:

"Both Sz and Bit Grooming algorithms seem valuable for compression in absolute error-bounded compression mode."

-page 11, line 9-10: I'd include characteristics of the data (e.g., maximum abs. value) earlier in the text when the two datasets are introduced.

Your suggestion will be taken into account: the characteristics of the data will be introduced in section 4.2.

We have added the characteristics of the data in section 4:

"Datasets $s1$ and $s3D$ were generated, $s1$ being a noisy sinusoid of 1 dimension with a maximum absolute value of 118. The data volume of the $s1$ dataset is 4MB. Dataset s3D is a noisy sinusoid pulse of 3 dimensions with a maximum absolute value of 145. The data volume of the $s3D$ dataset is 512MB."

-page 11, line 24: I don't see relative error mentioned in Table 6 - it seems to just be absolute error

The text will be modified to make it clearer: "…all three algorithms respect the maximum absolute error of 0.5 which, for the signal s1, corresponds to a relative error of 0.00424."

The text has been modified as follows:

"… all three algorithms respect the maximum absolute error of 0.5, which corresponds for dataset s1 to a relative error of 0.00424."

-p.11-12: Need more of a discussion of the results in Figure 7. For 3D, it looks like bit grooming and digit rounding are similar - I don't see a clear advantage.

More discussion on the results will be added to the text. For the s3D you are right, there is no clear advantage. It is written "the Digit Rounding algorithm provides compression performance very closed to the one of the Bit Grooming algorithm".

The text has been modified as follows:

"All three algorithms provide similar SNR versus compression ratio results, with a slight advantage for the Bit Grooming algorithm."

-p.12, lines 16-17: SZ compression can be controlled with an absolute error bound, so why is the relative error bound adjusted to get the desired abs. error?

The objective was to see if Sz compression configured with a relative error bound respect the error bound specified. As the digit rounding and bit-grooming algorithm can only be configured on a number

of significant digits, they can only "produce" absolute error in 0.5, 0.05, 0.005, etc. In order to be able to compare Sz configured with a relative error bound with those algorithms, we have configured the relative error bound to obtain a maximum absolute error of 0.5. These explanations will be added to the text.

The following sentence has been added in the text:

In order to be able to compare Sz configured with a relative error bound with those algorithms, we configured the relative error bound to obtain a maximum absolute error of 0.5: the pw_relBoundRatio parameter in Sz was set to 0.00424.

*-Section 5.1: It is disappointing not to have SZ results on the real data of interest. Were the SZ authors contacted? I would think that they could have helped resolve this issue.*

Yes, we had some exchanges. The issue is still under investigation.

A more recent version of Sz has been used and results on CFOSAT and SWOT datasets are now provided.

*-p. 15, line 18: "which only a few attributes may be missing" - It's unclear what this means. It's super helpful to really detail the data being compressed so that one can make sense of the results.*

Details on the datasets will be added to the text.

This part of the sentence has been removed has it is not relevant in the frame of this study. Details on the CFOSAT and SWOT datasets have been added in the Supplement, particularly the precision required for the compression of each variable.

*-p. 14, line 30: Please share more specific information about the precision required by the scientists for the data. Again, more information is useful for interpreting results.*

The configuration and the precision of each variable will be made available.

These details have been added in the Supplement.

*-Section 5 seems like it should be the highlight of the paper as here we are seeing the results on the real data. But it feels like more detail is needed on the data and more discussion of the implications of the results.*

Section 5 will be developed to add more details on the data and more discussion on the results obtained.

Section 5 has been reworked. It now provides results on particular variables of the CFOSAT and SWOT datasets. Details on the data have been added in the Supplement.

*——————————— Typos, etc.: ———————————*

*-abstract, line 7: incorrect use of "imposes"*

*-p.1, line 22: "quite spread"=> "quite prevalent" or "quite popular" , "widely spread" => "widely used"*

*-p.1., line 26: "reduce significantly" => "significantly reduce"*

*-p.1. line 27: This sentence (that continues to page 2) is too long.*

*-p.2, line 3: "can afford for" is awkward*

*-p2, lines 10-24: this region is 5 paragraphs*

*-p.3, lines 3-4: awkwardly worded*

*-p. 3, line 9: one sentence paragraph*

*-p.3, line 12: missing "," after "Deflate"*

*-p.3 line 14: not sure what is meant by "new concurrent"*

*-p.3, line 13: "widely spread" => "widely used"*

*-p.3, line 16: awkward sentence: "This allows Deflate achieving rather high compression ratios"*

*-P.3, section 3: again, there are too many tiny paragraphs*

*-p.4, line 7: "are of same interest" is awkward*

*-p.4, line 21: Table number is not given*

*-p.5, line 15: awkwardly worded*

*-p.5, line 24: One sentence paragraph*

*-p.7, section 4.2: define f_s, f_ech*

*-p.7, line 22: "use embarks" is awkward*

*-p.8, line 7: "declined" doesn't make sense*

*-p.8, line 14: "embark" - incorrect usage*

*-p.10, line 10: "This correspond corresponding" needs to be fixed*

*-p. 10, line 24: Note sure I'd use "performances" here as earlier it was used to indicate speed.*

*-p. 15, line 21: "ration" => "ratio"*

*-p.15, line 14: another one sentence paragraph*

*-p.16, line 9: "Extends to this work" - awkwardly worded*

Response: we thank you for highlighting typos that will help us to improve the manuscript.

All these points have been corrected.

**Reply to Zender (Referee)**

We are grateful to the referee for his constructive and thorough criticism and suggestions to our manuscript. Please find below a detailed point-by-point reply (*referee's comment in italic*).

*General Comments*

*This manuscript presents a new lossy compression algorithm called "Digit Rounding" (DR), and evaluates its performance against and with other lossy and lossless compression algorithms on idealized and remote sensing datasets. The manuscript addresses the growing need to archive meaningful data rather than noise, and to do so reliably and quickly. The study presents an original advance in lossy compression whose implementation unfortunately hampers its utility. The study is understandable yet poorly written. This potentially useful study of lossy compression techniques needs a thorough overhaul before publication.*

We will improve the writing of the manuscript by contacting a native English speaker/writer. As suggested by the Anonymous Referee #1, the paper will be reworked to highlight the main contributions of our work and focus more on the application to CFOSAT and SWOT datasets.

The grammar and writing have been improved throughout the manuscript and the paper has been reworked to highlight our main contribution and to focus more on the application to CFOSAT and SWOT datasets.

*Specific Comments*

*Originality: DR is an improvement on "Bit Grooming" (BG) which I invented as an improvement on "Bit Shaving". In that sense I am qualified to comment on its originality. The heart of DR is essentially a continuous version of BG: Whereas BG fixes the number of bits masked for each specified precision, and masks these bits for every value, DR recomputes the number of bits masked for each quantized value to achieve the same precision. BG did not implement the continuous method because I thought that computing the logarithm of each value would be expensive, inelegant, and yield only marginally more compression. However, DR cleverly uses the exponent field instead of computing logarithms, and so deciphers the correct number of bits to mask while avoiding expensive floating point math. This results in significantly more compressibility that (apparently) incurs no significant speed penalty (possibly because it compresses better and thus the lossless step is faster?). Hence DR appears to be a significant algorithmic advance and I congratulate the authors for their insight.*

Thank you for your congratulations. They are much appreciated. Indeed, the speed penalty of DR is compensated by the fact that the lossless step is faster.

In the previous version, we cascaded a call to ncks with a call to h5repack to perform Bit Grooming followed by Zstandard compression. For fairer comparisons on the compression speed, we modified our approach and now only employ ncks tool to run Decimal Rounding and Bit Grooming. However, we could not call Zstandard compression via the ncks tool, but only Deflate compression. Consequently, we now provide results for Decimal Rounding, Bit Grooming, Sz and Digit Rounding followed by Deflate compression.

*The manuscript stumbles in places due to low quality English, and cries out for more fluent editing. Not only is the word choice often awkward, but the manuscript is like a continuously choppy sea of standalone sentences with few well developed paragraphs that swell with meaning then yield gently to the next idea. GMD readers deserve and expect better.*

We will improve the grammar and writing (see first answer). We will also modify the paragraph structures to avoid too small paragraphs and better take care of the transitions.

The grammar and writing have been improved throughout the manuscript. Short paragraphs have also been modified and sentence transitions improved.

*Does DR guarantee that it will never create a relative error greater than half the value of the least significant digit? BG chooses the number of digits to mask conservatively, so it can and does guarantee that it always preserves the specified precision. Equations (1)-(7) imply that DR can make the same claim, but this claim is never explicitly tested or made. The absence of this guarantee is puzzling because it would strengthen the confidence of users in the algorithm. However, the guarantee must be explicitly tested, because it undergirds the premise that the comparison between DR and BG is fair. In any case, clearly state whether DR ever violates the desired precision, even if that happens only rarely.*

Equations (1)-(7) imply that DR guarantees that it always preserves the specified precision. We will explicitly add that claim in the text and show that DR always provides the desired precision on the number Pi with nsd varying from 1 to 8. We will also provide the maximum absolute error on artificial data of 1 000 000 values spanning [1.0, 2.0) in equal-increment steps of 1e-6.

We have added the following sentence below Eq. 4.

"This condition guarantees that the Digit Rounding algorithm to always preserves a relative error lower than or equal to half the value of the least significant digit."

We have also added results of DR algorithm on the number Pi in Table 2:

"Table 2 provides the result of the Digit Rounding algorithm on the value of $\pi$ with specified precisions $\mathrm{nsd}$ varying from 1 to 8 digits. It can be compared to the Bit Grooming results provided in Table 2 in (Zender, 2016a)."

We also provide the maximum absolute error on artificial data of 1 000 000 values spanning [1.0, 2.0) in equal-increment steps of 1e-6 in Table 3

"Table 3 provides the maximum absolute error obtained with varying $\mathrm{nsd}$ values on an artificial dataset composed of 1,000,000 values evenly spaced over the interval [1.0, 2.0). This is the same artificial dataset used in Table 3 in (Zender, 2016a). It shows that Digit Rounding always preserves a relative error lower than or equal to half the value of the least significant digit, i.e. $|s_i - \tilde{s}_i| \leq 0.5 \times 10^{d_i - \mathrm{nsd}}$."

*p. 16 L13: "Code and data availability: The Digit Rounding software source code and the data are currently only available upon request to Xavier Delaunay (xavier.delaunay@thalesgroup.com) or to Flavien Gouillon (Flavien.Gouillon@cnes.fr)." The GMD policy on code and data is here: https://www.geoscientific-model-development.net/about/code_and_data_policy.html. This manuscript provides no code access nor explanation, and no dataset access, and thus appears to violate GMD policy in these areas.*

The code and the datasets will be made publicly available on the CNES gitlab.

The code is now publicly available on CNES GitHub at https://github.com/CNES/Digit_Rounding and the dataset are available on demand.

"The Digit Rounding software source code is available from CNES GitHub at https://github.com/CNES/Digit_Rounding. The datasets are available upon request to Xavier Delaunay (xavier.delaunay@thalesgroup.com) or to Flavien Gouillon (Flavien.Gouillon@cnes.fr). The Supplement details the datasets and provides the command lines used for running the compression tools."

*Common comparisons would help build confidence in your results. It would have been more synergistic to evaluate the algorithms on at least one of the same datasets as Zender (2016), which are all publicly available. I am glad the authors used the publicly available NCO executables. Why not release the DR software in the same spirit so that the geoscience community can use (and possibly improve) it?*

Comparisons with BG will be provided on the same MERRA dataset used in Zender (2016). The DR software will be released under MIT-style open source license.

We have added results of DR on the same MERRA dataset used in Zender (2016).

"We compare the compression ratio obtained with the Digit Rounding algorithm to that obtained with the Bit Grooming algorithm for the same meteorological data from MERRA re-analysis studied in (Zender, 2016a). Table 4 reports the Bit Grooming results extracted from Table 6 in (Zender, 2016a) and provides the results of the Digit Rounding algorithm. The same lossless compression is employed: Shuffle and Deflate with level 1 compression. From $nsd = 7$ to $nsd = 5$, Digit Rounding and Bit Grooming provide similar compression ratios with a slight advantage for the Bit Grooming algorithm. However, from $nsd = 4$ to $nsd = 1$, the compression ratios obtained with Digit Rounding are clearly better."

The DR software is released under LGPL-v3 open source license.

*The lossless and lossy compression algorithms analyzed seem like a fairly balanced collection of those most relevant to GMD readers. Most methods that were omitted are, to my knowledge, either non-competitive (e.g., Packing) or not user-friendly, e.g., research grade but not widely available (e.g., Layer Packing) and too hard to independently implement.*

*Table 6 on p. 19 shows the maximum absolute error (MAE) of BG is quite similar to DR, as I would expect. However, Table 7 on p. 20 shows the maximum absolute error (MAE) of BG is nearly 10x less than DR. Why are the MAEs similar for dataset s1 and significantly different for dataset s3D? I expect DR has a greater mean error (and lower SNR) than BG due to the algorithms, yet the difference in MAEs surprises me. Zender (2016) Table 3 shows that BG is tuned to have an MAE just shy of violating the precision guarantee. An MAE that is nearly 10x larger seems like it might violate the precision guarantee.*

These results show that BG can sometimes be too conservative. As shown in Table 1 on the value Pi, BG sometimes preserves more bits in the mantissa than what is strictly necessary to achieve the required precision. This is what happens on the dataset s3D. On the contrary, DR adapts the quantization step to each value of the input dataset. Doing so, it can achieve the required precision while preserving less mantissa bits than DR does. This results both in a higher mean absolute error and in a higher MAE than BG. This explanation will be added to the text.

Thanks to you remark on the MAE on s1 dataset, it has been observed that DR algorithm was also too conservative on some values. It has been enhance in order to provide a MAE closer to what was expected. For this, the value $\log_{10}(m_i)$ is now tabulated with a few values.

"The $\log_{10}(m_i)$ value is tabulated. Only 5 tabulated values are used in our implementation, enough to provide a good precision. The tabulated v values for $\log_{10}(m_i)$ are such that $v \le \log_{10}(m_i)$. They are provided in the Supplement. This computation slightly underestimates the values for $d_i$ but provides a more conservative quantization, thus guaranteeing the specified number of significant digits."

The following sentence has also been added in the text:

"Bit Grooming is too conservative. It preserves more mantissa bits than strictly necessary to achieve the required precision. This behavior is illustrated in Table 1 with the value of π. In contrast, Digit Rounding adapts the quantization step to each value of the input dataset. Doing so, it can achieve the required precision while preserving less mantissa bits than Bit Grooming does. This results both in a higher maximal absolute error and in a higher mean absolute error than Bit Grooming, but also in a higher compression ratio."

*The preceding comment is a request to more carefully analyze the underlying cause of the behaviors reported in the data. The next two comments are to report more results to deepen the analyses and explain the behavior of DR more robustly.*

*Please include the maximum absolute error or maximum absolute relative error (which normalizes the error by the original value) to Tables 5–10.*

*MeanAE is an important statistic that is complementary to MaxAE. MeanAE is the average absolute (no compensation between positive and negative) bias in the dataset, and is more familiar and relevant than SNR to at least some geophysicists. Please consider including MeanAE in Tables 5–10.*

As suggested, the maximum absolute error and the mean absolute error (MeanAE) will be added to the tables allowing deeper analysis of DR behavior.

The maximum absolute error and the mean absolute error have been added to tables 5, 6 and 7. Tables 9, 12 and 13 provide compression results on CFOSAT and SWOT which are composed of several different datasets. The maximum absolute error and the mean absolute error could only be computed variable per variable. We thus now provide the results obtained on the ground_range_5 variable of the CFOSAT dataset in Table 8, the results obtained on the *height* variable of the SWOT dataset in Table 10, and the results obtained on the *pixel_area* variable of the other SWOT dataset in Table 11.

*Zender (2016) and Silver and Zender (2017) consider four primary criteria to evaluate compression algorithms: Compression Ratio, Accuracy, Speed, and User-friendliness. This manuscript neglects explicit consideration of the last, though usability seems (in addition to performance) seems to be an implicit reason why they recommend BG not DR for the "real world" use cases in Sections 5.1 and 5.2. The manuscript would benefit from a more explicit consideration of usability throughout. Examples include software availability, flexibility, and complexity of invocation, as well as transparency (will users have all the necessary software required to read the compressed data?), and instructions to mitigate these issues for DR.*

As for BG, there is no "decompression" associated to DR. DR does not require any software to read the rounded data. This argument will be added into the text. The reason why BG is recommended rather than DR for the compression of CFOSAT dataset in section 5.1 is that this dataset is compressed in absolute error bounded compression mode. DR only works for relative error bounded compression mode. Nevertheless, some results using DR on this dataset will be provided for completeness. In section 5.2, BG (in the absolute error bounded compression mode) is recommended

rather than DR for the compression SWOT L2 pixel cloud product. This recommendation is based on the compression ratio obtained. We will add the maximum absolute error and the mean absolute error (MeanAE) to Tables 8 to 10 for fairer comparisons. Moreover, we will provide a supplement to the article with the commands and datasets necessary to reproduce the results.

We have added the following sentences in the text:

"We have developed an HDF5 dynamically loaded filter plugin so as to apply the Digit Rounding algorithm to NetCDF-4 or HDF5 datasets. It should be noted that data values rounded by the Digit Rounding algorithm can be read directly: there is no reverse operation to Digit Rounding, and users do not need any software to read the rounded data."

Moreover, we have added some results using DR on the CFOSAT dataset for completeness.

The maximum absolute error and the mean absolute error have not been added to Tables 9, 12 and 13, because, as explained in the previous answer, CFOSAT and SWOT dataset are composed of several different variable.

We also now provide a supplement to the article with the commands and datasets necessary to reproduce the results.

*Tables 1 and 3 follow Tables 1 and 2 of Zender (2016). This should be noted in the text and/or caption of the tables.*

The reference to Zender (2016) will be added in the caption of Tables 1 and 3.

The captions have been modified as follows:

"Table 1: Representation of the value of π in IEEE-754 single-precision binary representation (first row) and results preserving 4 significant digits with the Bit Grooming algorithm (second row) or preserving 12 mantissa bits (third row). This table builds on Table 1 in (Zender, 2016a)."

"Table 2: Representation of the value of π in IEEE-754 single-precision binary representation (first row) and results preserving a varying number of significant digits (nsd) with the Digit Rounding algorithm. This table can be compared to Table 2 in (Zender, 2016a) providing the Bit Grooming results for π."

*It seems like Table 2, the algorithm description, should be a figure rather than a table.*

This will be corrected as suggested.

The algorithm description is now provided in Figure 2.

*The manuscript is awkward in that it introduces a demonstrably superior lossy compression algorithm but recommends a different algorithm (BG) for "real world" cases (Section 5), partly because DR is unavailable in software that potential users have easy access to, and its implementation appears to be too inflexible to use on generic datasets. The recommendation of BG not DR does attest to the objectivity of the study, yet it seems to be an unsatisfying conclusion to what was clearly a time-consuming study. In this sense the manuscript seems premature, since if DR were "ready for primetime" then the authors could have recommended it rather than BG in Section 5. Perhaps the authors should re-evaluate whether the manuscript is premature, i.e., whether it should both introduce*

*a new lossy algorithm before it is ready to use in optimized workflows for generic geoscientific data compression.*

As previously answered, the manuscript will be reworked to highlight the main contributions of our work and focus on the applications to the CFOSAT and the SWOT datasets. The maximum absolute error and the mean absolute error (MeanAE) will be added to Tables 5 to 10 for fairer comparisons that will allow mitigating the previous conclusions that were based on the compression ratio only. Moreover, some results using DR on CFOSAT dataset will be provided for completeness of the manuscript.

We have added some results using DR on the CFOSAT dataset for completeness, but also maximum and mean absolute error in the tables (see previous answers).

The conclusion has been reworked to make it clearer that we recommend Decimal Rounding for absolute error bounded compression of CFOSAT data but Digit Rounding for relative error bounded compression of SWOT data.

*Minor Suggestions*

*p. 1 L22: "well spread"*

*p. 2 L22: DEFLATE*

*p. 4 L1: maxi is redundant. Just use max.*

*p. 4 L21: Table 1*

*p. 9 L7: "declined"?*

*p. 9 L14: "By default, Sz algorithm embark Deflate." is awkward.*

*p. 14 L27–28: These lines are identical*

*p. 18 L8: "the number di of significant digit number of digits"???*

*p. 18 L8: "following Eq." not "following in Eq."*

*p. 23 Figure 4: Clarify the meaning of the distinct vertical bars.*

Response: we thank you for these suggestions that will help us to improve the manuscript.

All these points have been corrected.

Dear Editor,

Thank you so much for extending our time for the revision of the manuscript. We also like to thank both reviewers for their thorough review of the manuscript. Below, find the main modifications made to the manuscript which covers all the remarks.

- As suggested by the reviewers, the manuscript has been reworked to focus more on the application to CFOSAT and SWOT datasets. We have also moved the details concerning the synthetic datasets in the Supplement which is now provided with the paper.

- The term "Bit Grooming" was employed both for the absolute and the relative error bounded compression modes. But the algorithms are different. In the revised version of the paper we employ the term "Decimal Rounding" in the absolute error bounded mode and the term "Bit Grooming" in the relative absolute error bounded mode to remove any ambiguities.

- Thanks to a remark from C. Zender, we found that the Digit Rounding algorithm was sometimes too conservative. We thus slightly modified the implementation. The compression results reported in the new version of the paper have been obtained with the new version of the Digit Rounding algorithm.

- In the previous version of the paper, we cascaded a call to ncks with a call to h5repack to perform Bit Grooming followed by Zstandard compression. For fairer comparisons on the compression speed, we modified our approach and now only employ ncks tool to run Decimal Rounding and Bit Grooming. However, we could not call Zstandard compression via the ncks tool, but only Deflate compression. Consequently, we now provide results for Decimal Rounding, Bit Grooming, Sz and Digit Rounding followed by Deflate compression.

- The Digit Rounding software source code is now available from CNES GitHub at https://github.com/CNES/Digit_Rounding.

- We now provide a Supplement which details the datasets and provides the command lines used for running the compression tools.

- The grammar and English of the paper has been dramatically improved by reviews from native English speakers.

**Reply to Anonymous Referee #1**

We are grateful to the referee for her/his constructive and thorough criticism and suggestions to our manuscript. Please, find below a detailed point-by-point reply. *Referee's comments are in blue italic*; our answers are in black and our changes to the manuscript in green.

*- This manuscript needs a lot of improvement in terms of the grammar and writing. There are many awkward phrases and incorrect word choices that need to be improved (a subset are listed below). The paragraph structures are also in need of modification (many paragraphs contain only 1 or 2 sentences).*

We will improve grammar and writing of the manuscript by contacting a native English speaker/writer. Thank you for pointing out the subset of incorrect word choices. We will also modify the paragraph structures to avoid too small paragraphs.

The grammar and writing have been improved throughout the manuscript. Short paragraphs have also been modified.

*-Section 2: I'd be helpful to include more detail for the preprocessing algorithms: shuffle and bitshuffle. Also this section in general needs improvement. It's a bit "choppy" to read (needs smoother and better transitions between topics) and feels like more details would be helpful on the methods (especially the ones that the digit rounding algorithm builds on).*

More details will be added on the shuffle and bitshuffle algorithms. More details will also be added on the bit-grooming and decimal rounding algorithm, algorithm on which the digit rounding algorithm is built. We will do what is needed to improve this section in general.

Section 2 has been restructured so as to make the reading smoother. Details have been added on the shuffle and bitshuffle algorithms as well as on the bit-grooming and decimal rounding algorithm.

"The Shuffle algorithm groups all the data samples' first bytes together, all the second bytes together, etc. In smooth datasets, or datasets with highly correlated consecutive sample values, this rearrangement creates long runs of similar bytes, improving the dataset's compression. Bitshuffle extends the concept of Shuffle to the bit level by grouping together all the data samples' first bits, second bits, etc.

…

The Decimal Rounding algorithm achieves a uniform scalar quantization of the data. The quantization step is a power of 2 pre-computed so as to preserve a specific number of decimal digits. The Bit Grooming algorithm creates a bitmask to degrade the least significant bits of the mantissa of IEEE 754 floating-point data. Given a specified total number of significant digits, $nsd$, the Bit Grooming algorithm tabulates the number of mantissa bits that has to be preserved to guarantee the specified precision of $nsd$ digits: to guarantee 1-6 digits of precision, Bit Grooming must retain 5, 8, 11, 15, 18, and 21 mantissa bits respectively. The advantage is that the number of mantissa bits that must be preserved is computed very quickly. The disadvantage is that this computation is not optimal. In many cases, more mantissa bits are preserved than strictly necessary."

*-Section 4: Why does using the synthetic data in 4.1 to assess performance make sense - it seems unrelated to the application area of interest. I'd argue that the metrics used in 4.1 are really minimal requirements as well. Also take care when referring to "performance" as it is overloaded term...do you mean speed or effectiveness (it's used both ways)*

The objective of using synthetic data was to control the data parameters, such as the SNR, to be able to assess the impact of these parameters on the compression ratios. The results are not reported in this paper which rather focuses on providing a comparison of the compression ratio and speed of different algorithms. It has also been chosen to present only the minimal set of relevant metrics to avoid overloading the paper. We will be more rigorous and replace the term "performance" by "compression ratio" or "compression speed" in the text.

All the occurrences of the term "performance" have been checked and corrected when needed. The Mean Absolute Error metric has been added to the list of metrics:

"mean absolute error $\overline{e_{abs}}$ to evaluate the mean data degradation. It is defined as the mean of the pointwise absolute difference between the original and compressed data:

$$\overline{e_{abs}} = \frac{1}{N}\sum_{i=0}^{N-1}|s_i - \tilde{s}_i|"$$

*-fpzip is a fast and effective lossless method that would have been nice to compare (I \*think\* there is an fpzip filter available). Also I believe that any hdf5 filter can be accessed through NetCDF4 (see last sentence in conclusion) - consider contacting the Unidata folks.*

Thank you for the suggestion. Indeed a HDF5 filter is accessible for fpzip. However, many lossless compression algorithms exist. In our paper, we chose to evaluate the most "popular", i.e. the lossless compression algorithms the most used in applications. Thank you for pointing this evolution of the NetCDF-4 library: from version 4.6.0 - January 24, 2018, NetCDF fully supports HDF5 dynamic filters. The text of the paper will be modified so as to provide the example usage using the new NetCDF-4 features.

An example using the new NetCDF-4 features with *nccopy* tool has been provided in the supplement and the conclusion has been modified to remove the last sentence. However, nccopy tool does not allow yet linking together different filters.

*-Comments on doing compression in parallel?*

We do not consider running compression algorithm in parallel in this work and will make it clear in the manuscript. It is a possible extension of this study.

The following sentence has been added in section 4:

"Parallel compression has not been considered in this work."

*-When reading the conclusion, it's hard to see what the main contributions of this paper are. It's fairly well known already that preprocessing of scientific data (e.g., bit shuffle or shuffle) improves lossy compression. Also the statements in the conclusion aren't specific to a particular type of data set, but are presented as more general conclusions.*

*Given that the effectiveness and performance (speed) of lossy and lossless compression are very data, application, and variable dependent, the general statements here are not well justified by the small sample of data in the paper. I'd suggest focusing the paper more heavily on the data in Section 5 (if it's of interest) and tailoring the discussion in that manner. Or maybe the focus was to be more on speeds than quality, in which case it's be important to work to get sz and fpzip working, particularly via netcdf-4...*

Thank you for the suggestion that will help highlighting the main contributions of our work. As suggested, the paper will be reworked to focus more on the application to the CFOSAT and SWOT datasets. We will also avoid general statements but attach our conclusions to our application case.

The paper has been reworked to focus more on the application to the CFOSAT and SWOT datasets. Sz compression has been run on CFOSAT dataset and on some parts of SWOT datasets. The NetCDF-4 tool nccopy does not allow yet linking together different filters. This restrains its usability and this is why we prefer using h5repack tool.

————————— Specific items: —————————-

-p2, line 20: note that fpzip can also be lossless

Thank you for the remark. Fpzip will be presented both as a lossless and lossy compression algorithm.

The text has been modified as follows:

"Third, some lossy/lossless compression algorithms, such as FPZIP (Lindstrom and Isenburg, 2006), …"

-p. 8: discussion of figure 5: is the width of the bars related to the compression levels? (e.g. line 20 statement is unclear)

No. All the bars have the same width. Each vertical bar represents a compression level. For instance, the 9 compression levels of Deflate are represented by 9 vertical bars. This will be clarified in the text p.8.

Clarifications have been added to the text:

"The vertical bars represent the results for different compression levels: from 1 to 9 for the Deflate level dfl_lvl, from 1 to 22 for Zstandard level zstd_lvl, and only one level for LZ4."

-p.8, lines 28-29: Why is this the case? (Add some discussion beyond describing the figure.)

These lower compression/decompression speeds are not well understood and would require further investigation to be fully understood. It might be related to HDF5 chunking. Indeed, HDF5 split the data into chunks of small size that are independently compressed. This allows HDF5 to improve partial I/O for big datasets but can sometimes reduce the compression/decompression speeds. This discussion will be added to the text.

The following sentence has been added to the text:

"Further investigations are required to understand why the compression/decompression speeds are lower, but it might be related to HDF5 chunking."

*-p.8, line 20: I feel like the parameters should be better explained for –filter so that the reader can try them more easily. For example, what does the "32017" mean? I think that the following 0 is for sz, but this is not stated either.*

We will add the meaning of each parameters. Each HDF5 filter is identified by a unique ID. "32017" is the identifier of Sz filter. The following "0" is the number of filter parameters. In the case of Sz, the filter does not have any parameter to set. That is why there are 0 parameters. Sz compressor is configured via the sz.config file. The same explanations will be added for the other filters used in the paper.

All the command lines have been moved to the Supplement with the explanation above to avoid overloading the manuscript.

*-p. 10, line 6: re: "experiments have shown" - whose or which experiments (cite?)*

It is based on our own experiments that haven't been published. The sentence will be reworked as follows: "We have found that Shuffle or Bitshuffle preprocessing do not increase the compression ratio when applied after Sz. We have also found that and Bitshuffle provide lower compression ratio than Shuffle when applied after Bit Grooming. That is why only Shuffle is applied after Bit Grooming."

This sentence has been removed since the results have not been published.

*-Table 5 : Why is the speed faster for 1d?*

As previously, the lower compression/decompression speeds obtained with the dataset s3D are not well understood and might be related to HDF5 chunking. This discussion will be added to the text.

The following sentence has been added to the text:

"The lower compression/decompression speeds obtained with Sz on the dataset s3D are not well understood and might be related to HDF5 chunking as previously mentioned."

*-Section 4.4.1, line 21-24: Any idea why you get these results?*

Sz performs better on smooth signals since it makes use of a prediction step. The signal s1 being highly noisy, Sz prediction might often fail. This can explain the lower compression ratio on the signal s1. On the contrary, Bit-grooming does not makes any prediction. This can explain why it achieves better compression than Sz on the signal s1. This hypothesis will be added to the text.

The following sentences have been added to the text:

"Sz may perform better on dataset $s3D$ because it is smoother than dataset $s1$. Indeed, Sz integrates a prediction step. This prediction might often fail because dataset $s1$ is very noisy. This may explain the lower compression ratio for this dataset. Decimal Rounding, however, does not make any predictions, which may explain why it achieves a better compression than Sz for dataset $s1$."

*-Section 4.4.1, last sentences: It's unclear to me what the value of these synthetic data sets is - especially given the statement on p. 11, line 3, about the dependence on the dataset*

As suggested previously, the paper will be reworked to focus more on the application to CFOSAT and SWOT datasets without drawing general conclusions based on the results obtained on the synthetic datasets.

We modified the last sentence of this section as follows:

"Both Sz and Bit Grooming algorithms seem valuable for compression in absolute error-bounded compression mode."

*-page 11, line 9-10: I'd include characteristics of the data (e.g., maximum abs. value) earlier in the text when the two datasets are introduced.*

Your suggestion will be taken into account: the characteristics of the data will be introduced in section 4.2.

We have added the characteristics of the data in section 4:

"Datasets $s1$ and $s3D$ were generated, $s1$ being a noisy sinusoid of 1 dimension with a maximum absolute value of 118. The data volume of the $s1$ dataset is 4MB. Dataset s3D is a noisy sinusoid pulse of 3 dimensions with a maximum absolute value of 145. The data volume of the $s3D$ dataset is 512MB."

*-page 11, line 24: I don't see relative error mentioned in Table 6 - it seems to just be absolute error*

The text will be modified to make it clearer: "…all three algorithms respect the maximum absolute error of 0.5 which, for the signal s1, corresponds to a relative error of 0.00424."

The text has been modified as follows:

"… all three algorithms respect the maximum absolute error of 0.5, which corresponds for dataset s1 to a relative error of 0.00424."

*-p.11-12: Need more of a discussion of the results in Figure 7. For 3D, it looks like bit grooming and digit rounding are similar - I don't see a clear advantage.*

More discussion on the results will be added to the text. For the s3D you are right, there is no clear advantage. It is written "the Digit Rounding algorithm provides compression performance very closed to the one of the Bit Grooming algorithm".

The text has been modified as follows:

"All three algorithms provide similar SNR versus compression ratio results, with a slight advantage for the Bit Grooming algorithm."

*-p.12, lines 16-17: SZ compression can be controlled with an absolute error bound, so why is the relative error bound adjusted to get the desired abs. error?*

The objective was to see if Sz compression configured with a relative error bound respect the error bound specified. As the digit rounding and bit-grooming algorithm can only be configured on a number

of significant digits, they can only "produce" absolute error in 0.5, 0.05, 0.005, etc. In order to be able to compare Sz configured with a relative error bound with those algorithms, we have configured the relative error bound to obtain a maximum absolute error of 0.5. These explanations will be added to the text.

The following sentence has been added in the text:

In order to be able to compare Sz configured with a relative error bound with those algorithms, we configured the relative error bound to obtain a maximum absolute error of 0.5: the pw_relBoundRatio parameter in Sz was set to 0.00424.

*-Section 5.1: It is disappointing not to have SZ results on the real data of interest. Were the SZ authors contacted? I would think that they could have helped resolve this issue.*

Yes, we had some exchanges. The issue is still under investigation.

A more recent version of Sz has been used and results on CFOSAT and SWOT datasets are now provided.

*-p. 15, line 18: "which only a few attributes may be missing" - It's unclear what this means. It's super helpful to really detail the data being compressed so that one can make sense of the results.*

Details on the datasets will be added to the text.

This part of the sentence has been removed has it is not relevant in the frame of this study. Details on the CFOSAT and SWOT datasets have been added in the Supplement, particularly the precision required for the compression of each variable.

*-p. 14, line 30: Please share more specific information about the precision required by the scientists for the data. Again, more information is useful for interpreting results.*

The configuration and the precision of each variable will be made available.

These details have been added in the Supplement.

*-Section 5 seems like it should be the highlight of the paper as here we are seeing the results on the real data. But it feels like more detail is needed on the data and more discussion of the implications of the results.*

Section 5 will be developed to add more details on the data and more discussion on the results obtained.

Section 5 has been reworked. It now provides results on particular variables of the CFOSAT and SWOT datasets. Details on the data have been added in the Supplement.

*———————————— Typos, etc.: ————————————*

*-abstract, line 7: incorrect use of "imposes"*

*-p.1, line 22: "quite spread"=> "quite prevalent" or "quite popular" , "widely spread" => "widely used"*

*-p.1., line 26: "reduce significantly" => "significantly reduce"*

*-p.1. line 27: This sentence (that continues to page 2) is too long.*

*-p.2, line 3: "can afford for" is awkward*

*-p2, lines 10-24: this region is 5 paragraphs*

*-p.3, lines 3-4: awkwardly worded*

*-p. 3, line 9: one sentence paragraph*

*-p.3, line 12: missing "," after "Deflate"*

*-p.3 line 14: not sure what is meant by "new concurrent"*

*-p.3, line 13: "widely spread" => "widely used"*

*-p.3, line 16: awkward sentence: "This allows Deflate achieving rather high compression ratios"*

*-P.3, section 3: again, there are too many tiny paragraphs*

*-p.4, line 7: "are of same interest" is awkward*

*-p.4, line 21: Table number is not given*

*-p.5, line 15: awkwardly worded*

*-p.5, line 24: One sentence paragraph*

*-p.7, section 4.2: define f_s, f_ech*

*-p.7, line 22: "use embarks" is awkward*

*-p.8, line 7: "declined" doesn't make sense*

*-p.8, line 14: "embark" - incorrect usage*

*-p.10, line 10: "This correspond corresponding" needs to be fixed*

*-p. 10, line 24: Note sure I'd use "performances" here as earlier it was used to indicate speed.*

*-p. 15, line 21: "ration" => "ratio"*

*-p.15, line 14: another one sentence paragraph*

*-p.16, line 9: "Extends to this work" - awkwardly worded*

Response: we thank you for highlighting typos that will help us to improve the manuscript.

All these points have been corrected.

**Reply to Zender (Referee)**

We are grateful to the referee for his constructive and thorough criticism and suggestions to our manuscript. Please find below a detailed point-by-point reply (*referee's comment in italic*).

*General Comments*

*This manuscript presents a new lossy compression algorithm called "Digit Rounding" (DR), and evaluates its performance against and with other lossy and lossless compression algorithms on idealized and remote sensing datasets. The manuscript addresses the growing need to archive meaningful data rather than noise, and to do so reliably and quickly. The study presents an original advance in lossy compression whose implementation unfortunately hampers its utility. The study is understandable yet poorly written. This potentially useful study of lossy compression techniques needs a thorough overhaul before publication.*

We will improve the writing of the manuscript by contacting a native English speaker/writer. As suggested by the Anonymous Referee #1, the paper will be reworked to highlight the main contributions of our work and focus more on the application to CFOSAT and SWOT datasets.

The grammar and writing have been improved throughout the manuscript and the paper has been reworked to highlight our main contribution and to focus more on the application to CFOSAT and SWOT datasets.

*Specific Comments*

*Originality: DR is an improvement on "Bit Grooming" (BG) which I invented as an improvement on "Bit Shaving". In that sense I am qualified to comment on its originality. The heart of DR is essentially a continuous version of BG: Whereas BG fixes the number of bits masked for each specified precision, and masks these bits for every value, DR recomputes the number of bits masked for each quantized value to achieve the same precision. BG did not implement the continuous method because I thought that computing the logarithm of each value would be expensive, inelegant, and yield only marginally more compression. However, DR cleverly uses the exponent field instead of computing logarithms, and so deciphers the correct number of bits to mask while avoiding expensive floating point math. This results in significantly more compressibility that (apparently) incurs no significant speed penalty (possibly because it compresses better and thus the lossless step is faster?). Hence DR appears to be a significant algorithmic advance and I congratulate the authors for their insight.*

Thank you for your congratulations. They are much appreciated. Indeed, the speed penalty of DR is compensated by the fact that the lossless step is faster.

In the previous version, we cascaded a call to ncks with a call to h5repack to perform Bit Grooming followed by Zstandard compression. For fairer comparisons on the compression speed, we modified our approach and now only employ ncks tool to run Decimal Rounding and Bit Grooming. However, we could not call Zstandard compression via the ncks tool, but only Deflate compression. Consequently, we now provide results for Decimal Rounding, Bit Grooming, Sz and Digit Rounding followed by Deflate compression.

*The manuscript stumbles in places due to low quality English, and cries out for more fluent editing. Not only is the word choice often awkward, but the manuscript is like a continuously choppy sea of standalone sentences with few well developed paragraphs that swell with meaning then yield gently to the next idea. GMD readers deserve and expect better.*

We will improve the grammar and writing (see first answer). We will also modify the paragraph structures to avoid too small paragraphs and better take care of the transitions.

The grammar and writing have been improved throughout the manuscript. Short paragraphs have also been modified and sentence transitions improved.

*Does DR guarantee that it will never create a relative error greater than half the value of the least significant digit? BG chooses the number of digits to mask conservatively, so it can and does guarantee that it always preserves the specified precision. Equations (1)-(7) imply that DR can make the same claim, but this claim is never explicitly tested or made. The absence of this guarantee is puzzling because it would strengthen the confidence of users in the algorithm. However, the guarantee must be explicitly tested, because it undergirds the premise that the comparison between DR and BG is fair. In any case, clearly state whether DR ever violates the desired precision, even if that happens only rarely.*

Equations (1)-(7) imply that DR guarantees that it always preserves the specified precision. We will explicitly add that claim in the text and show that DR always provides the desired precision on the number Pi with nsd varying from 1 to 8. We will also provide the maximum absolute error on artificial data of 1 000 000 values spanning [1.0, 2.0) in equal-increment steps of 1e-6.

We have added the following sentence below Eq. 4.

"This condition guarantees that the Digit Rounding algorithm to always preserves a relative error lower than or equal to half the value of the least significant digit."

We have also added results of DR algorithm on the number Pi in Table 2:

"Table 2 provides the result of the Digit Rounding algorithm on the value of $\pi$ with specified precisions $\mathrm{nsd}$ varying from 1 to 8 digits. It can be compared to the Bit Grooming results provided in Table 2 in (Zender, 2016a)."

We also provide the maximum absolute error on artificial data of 1 000 000 values spanning [1.0, 2.0) in equal-increment steps of 1e-6 in Table 3

"Table 3 provides the maximum absolute error obtained with varying $\mathrm{nsd}$ values on an artificial dataset composed of 1,000,000 values evenly spaced over the interval [1.0, 2.0). This is the same artificial dataset used in Table 3 in (Zender, 2016a). It shows that Digit Rounding always preserves a relative error lower than or equal to half the value of the least significant digit, i.e. $|s_i - \tilde{s}_i| \leq 0.5 \times 10^{d_i - \mathrm{nsd}}$. "

*p. 16 L13: "Code and data availability: The Digit Rounding software source code and the data are currently only available upon request to Xavier Delaunay (xavier.delaunay@thalesgroup.com) or to Flavien Gouillon (Flavien.Gouillon@cnes.fr)." The GMD policy on code and data is here: https://www.geoscientific-model-development.net/about/code_and_data_policy.html. This manuscript provides no code access nor explanation, and no dataset access, and thus appears to violate GMD policy in these areas.*

The code and the datasets will be made publicly available on the CNES gitlab.

The code is now publicly available on CNES GitHub at https://github.com/CNES/Digit_Rounding and the dataset are available on demand.

"The Digit Rounding software source code is available from CNES GitHub at https://github.com/CNES/Digit_Rounding. The datasets are available upon request to Xavier Delaunay (xavier.delaunay@thalesgroup.com) or to Flavien Gouillon (Flavien.Gouillon@cnes.fr). The Supplement details the datasets and provides the command lines used for running the compression tools."

*Common comparisons would help build confidence in your results. It would have been more synergistic to evaluate the algorithms on at least one of the same datasets as Zender (2016), which are all publicly available. I am glad the authors used the publicly available NCO executables. Why not release the DR software in the same spirit so that the geoscience community can use (and possibly improve) it?*

Comparisons with BG will be provided on the same MERRA dataset used in Zender (2016). The DR software will be released under MIT-style open source license.

We have added results of DR on the same MERRA dataset used in Zender (2016).

"We compare the compression ratio obtained with the Digit Rounding algorithm to that obtained with the Bit Grooming algorithm for the same meteorological data from MERRA re-analysis studied in (Zender, 2016a). Table 4 reports the Bit Grooming results extracted from Table 6 in (Zender, 2016a) and provides the results of the Digit Rounding algorithm. The same lossless compression is employed: Shuffle and Deflate with level 1 compression. From $\text{nsd} = 7$ to $\text{nsd} = 5$, Digit Rounding and Bit Grooming provide similar compression ratios with a slight advantage for the Bit Grooming algorithm. However, from $\text{nsd} = 4$ to $\text{nsd} = 1$, the compression ratios obtained with Digit Rounding are clearly better."

The DR software is released under LGPL-v3 open source license.

*The lossless and lossy compression algorithms analyzed seem like a fairly balanced collection of those most relevant to GMD readers. Most methods that were omitted are, to my knowledge, either non-competitive (e.g., Packing) or not user-friendly, e.g., research grade but not widely available (e.g., Layer Packing) and too hard to independently implement.*

*Table 6 on p. 19 shows the maximum absolute error (MAE) of BG is quite similar to DR, as I would expect. However, Table 7 on p. 20 shows the maximum absolute error (MAE) of BG is nearly 10x less than DR. Why are the MAEs similar for dataset s1 and significantly different for dataset s3D? I expect DR has a greater mean error (and lower SNR) than BG due to the algorithms, yet the difference in MAEs surprises me. Zender (2016) Table 3 shows that BG is tuned to have an MAE just shy of violating the precision guarantee. An MAE that is nearly 10x larger seems like it might violate the precision guarantee.*

These results show that BG can sometimes be too conservative. As shown in Table 1 on the value Pi, BG sometimes preserves more bits in the mantissa than what is strictly necessary to achieve the required precision. This is what happens on the dataset s3D.  On the contrary, DR adapts the quantization step to each value of the input dataset. Doing so, it can achieve the required precision while preserving less mantissa bits than DR does. This results both in a higher mean absolute error and in a higher MAE than BG. This explanation will be added to the text.

Thanks to you remark on the MAE on s1 dataset, it has been observed that DR algorithm was also too conservative on some values. It has been enhance in order to provide a MAE closer to what was expected. For this, the value $\log_{10}(m_i)$ is now tabulated with a few values.

"The $\log_{10}(m_i)$ value is tabulated. Only 5 tabulated values are used in our implementation, enough to provide a good precision. The tabulated v values for $\log_{10}(m_i)$ are such that $v \leq \log_{10}(m_i)$. They are provided in the Supplement. This computation slightly underestimates the values for $d_i$ but provides a more conservative quantization, thus guaranteeing the specified number of significant digits."

The following sentence has also been added in the text:

"Bit Grooming is too conservative. It preserves more mantissa bits than strictly necessary to achieve the required precision. This behavior is illustrated in Table 1 with the value of π. In contrast, Digit Rounding adapts the quantization step to each value of the input dataset. Doing so, it can achieve the required precision while preserving less mantissa bits than Bit Grooming does. This results both in a higher maximal absolute error and in a higher mean absolute error than Bit Grooming, but also in a higher compression ratio."

*The preceding comment is a request to more carefully analyze the underlying cause of the behaviors reported in the data. The next two comments are to report more results to deepen the analyses and explain the behavior of DR more robustly.*

*Please include the maximum absolute error or maximum absolute relative error (which normalizes the error by the original value) to Tables 5–10.*

*MeanAE is an important statistic that is complementary to MaxAE. MeanAE is the average absolute (no compensation between positive and negative) bias in the dataset, and is more familiar and relevant than SNR to at least some geophysicists. Please consider including MeanAE in Tables 5–10.*

As suggested, the maximum absolute error and the mean absolute error (MeanAE) will be added to the tables allowing deeper analysis of DR behavior.

The maximum absolute error and the mean absolute error have been added to tables 5, 6 and 7. Tables 9, 12 and 13 provide compression results on CFOSAT and SWOT which are composed of several different datasets. The maximum absolute error and the mean absolute error could only be computed variable per variable. We thus now provide the results obtained on the ground_range_5 variable of the CFOSAT dataset in Table 8, the results obtained on the *height* variable of the SWOT dataset in Table 10, and the results obtained on the *pixel_area* variable of the other SWOT dataset in Table 11.

*Zender (2016) and Silver and Zender (2017) consider four primary criteria to evaluate compression algorithms: Compression Ratio, Accuracy, Speed, and User-friendliness. This manuscript neglects explicit consideration of the last, though usability seems (in addition to performance) seems to be an implicit reason why they recommend BG not DR for the "real world" use cases in Sections 5.1 and 5.2. The manuscript would benefit from a more explicit consideration of usability throughout. Examples include software availability, flexibility, and complexity of invocation, as well as transparency (will users have all the necessary software required to read the compressed data?), and instructions to mitigate these issues for DR.*

As for BG, there is no "decompression" associated to DR. DR does not require any software to read the rounded data. This argument will be added into the text. The reason why BG is recommended rather than DR for the compression of CFOSAT dataset in section 5.1 is that this dataset is compressed in absolute error bounded compression mode. DR only works for relative error bounded compression mode. Nevertheless, some results using DR on this dataset will be provided for completeness. In section 5.2, BG (in the absolute error bounded compression mode) is recommended

rather than DR for the compression SWOT L2 pixel cloud product. This recommendation is based on the compression ratio obtained. We will add the maximum absolute error and the mean absolute error (MeanAE) to Tables 8 to 10 for fairer comparisons. Moreover, we will provide a supplement to the article with the commands and datasets necessary to reproduce the results.

We have added the following sentences in the text:

"We have developed an HDF5 dynamically loaded filter plugin so as to apply the Digit Rounding algorithm to NetCDF-4 or HDF5 datasets. It should be noted that data values rounded by the Digit Rounding algorithm can be read directly: there is no reverse operation to Digit Rounding, and users do not need any software to read the rounded data."

Moreover, we have added some results using DR on the CFOSAT dataset for completeness.

The maximum absolute error and the mean absolute error have not been added to Tables 9, 12 and 13, because, as explained in the previous answer, CFOSAT and SWOT dataset are composed of several different variable.

We also now provide a supplement to the article with the commands and datasets necessary to reproduce the results.

*Tables 1 and 3 follow Tables 1 and 2 of Zender (2016). This should be noted in the text and/or caption of the tables.*

The reference to Zender (2016) will be added in the caption of Tables 1 and 3.

The captions have been modified as follows:

"Table 1: Representation of the value of π in IEEE-754 single-precision binary representation (first row) and results preserving 4 significant digits with the Bit Grooming algorithm (second row) or preserving 12 mantissa bits (third row). This table builds on Table 1 in (Zender, 2016a)."

"Table 2: Representation of the value of π in IEEE-754 single-precision binary representation (first row) and results preserving a varying number of significant digits (nsd) with the Digit Rounding algorithm. This table can be compared to Table 2 in (Zender, 2016a) providing the Bit Grooming results for π."

*It seems like Table 2, the algorithm description, should be a figure rather than a table.*

This will be corrected as suggested.

The algorithm description is now provided in Figure 2.

*The manuscript is awkward in that it introduces a demonstrably superior lossy compression algorithm but recommends a different algorithm (BG) for "real world" cases (Section 5), partly because DR is unavailable in software that potential users have easy access to, and its implementation appears to be too inflexible to use on generic datasets. The recommendation of BG not DR does attest to the objectivity of the study, yet it seems to be an unsatisfying conclusion to what was clearly a time-consuming study. In this sense the manuscript seems premature, since if DR were "ready for primetime" then the authors could have recommended it rather than BG in Section 5. Perhaps the authors should re-evaluate whether the manuscript is premature, i.e., whether it should both introduce*

*a new lossy algorithm before it is ready to use in optimized workflows for generic geoscientific data compression.*

As previously answered, the manuscript will be reworked to highlight the main contributions of our work and focus on the applications to the CFOSAT and the SWOT datasets. The maximum absolute error and the mean absolute error (MeanAE) will be added to Tables 5 to 10 for fairer comparisons that will allow mitigating the previous conclusions that were based on the compression ratio only. Moreover, some results using DR on CFOSAT dataset will be provided for completeness of the manuscript.

We have added some results using DR on the CFOSAT dataset for completeness, but also maximum and mean absolute error in the tables (see previous answers).

The conclusion has been reworked to make it clearer that we recommend Decimal Rounding for absolute error bounded compression of CFOSAT data but Digit Rounding for relative error bounded compression of SWOT data.

*Minor Suggestions*

*p. 1 L22: "well spread"*

*p. 2 L22: DEFLATE*

*p. 4 L1: maxi is redundant. Just use max.*

*p. 4 L21: Table 1*

*p. 9 L7: "declined"?*

*p. 9 L14: "By default, Sz algorithm embark Deflate." is awkward.*

*p. 14 L27–28: These lines are identical*

*p. 18 L8: "the number di of significant digit number of digits"???*

*p. 18 L8: "following Eq." not "following in Eq."*

*p. 23 Figure 4: Clarify the meaning of the distinct vertical bars.*

Response: we thank you for these suggestions that will help us to improve the manuscript.

All these points have been corrected.

[revised manuscript text omitted]

10  the compression respectively by ratio by more20% and than30% with similar compression speeds and faster decompression. , but divides the compression speed by more than 3 the compression speed. The decompression speeds are similar for all the solutions tested. Our recommendation for the compression of this dataset is thus to use of Shuffle and Zstandard in lossless mode to achieve a very high compression speed, or either the Bbit bit grooming Grooming or the Digit Rounding algorithm to achieve a slightly higher compression ratio at the price of a lower compression speed.

15  The results for the compression of the representative SWOT L2 pixel cloud product are provided in Table 130.

 Compared to Deflatethe baseline, Zstandard compression is nearly 4 times fasterincreases by more than 2.5 times the compression speed while offering a similar compression ratio. The use of the Bit Grooming increases the compression ratio by 29% with higher compression speed. And Digit Rounding increases the compression ratio by 34% with slightly lower compression speed than Bit Grooming.  Bit Grooming and Digit Rounding provides the Bit Grooming algorithm in the

20  absolute error bounded mode increases more than 2 times the compression ratio by over twice but reduces the compression speed. The compression ratios obtained in the relative error bounded mode, either with the Bit Grooming algorithm or the Digit Rounding algorithms, are not as high. Thefastest decompression decompression speeds are similar for all the solutions tested. Our recommendation for the compression of this datasetSWOT datasets is thus to use the Bit GroomingDigit Rounding algorithm in the absolute error bounded mode to achieve high compression, at the price of a lower compression

25  speed than the lossless solutions, considering that for SWOT the driver is product size is a driver, and considering taking into account the ratioration between compression time and processing time.

**76 Conclusions**

We have studiedThis study investigatedevaluated the performance of lossless and lossy compression algorithms both on synthetic datasets and on realistic simulated datasets of future sciencetifiescientific satellites. The compression methods have

30  beenwere executed applied using NetCDF-4 and HDF5 tools.

 It has been shown that for the lossless compression of scientific datasets,dataset the compression performance is increased when preprocessing by Shuffle orof Bitshuffle are used for preprocessingis very helpful to increase the compression

performance. the compression level options of Zstandard or Deflate  on the compression ratio achieved is not significant compared to the impact of the Shuffle or Bitshuffle preprocessing. However, high compression levels  can significantly reduce the compression speed. ~~Low compression levels are thus a good choices if the goal is to achieve a high compression speed with a satisfactory compression ratio. Zstandard can provide similar as higher a compression speed than as 
[revised manuscript text omitted]

---

## Author Response (AR2)

Dear Editor,

We have included in this document the review Charlie Zender sent by e-mail. We thank the three referees for the time spent on reviewing this manuscript again and for their suggestions. Please, find below a detailed point-by-point reply. *Referee's comments are in blue italic*; our answers are in black and our changes to the manuscript in green.

**Reply to Anonymous Referee #3 (Report #1)**

We are grateful to the referee for her/his constructive and thorough criticism and suggestions to our manuscript.

The paper studied the performance of a series of lossless and lossy compression methods/filters that can be plugged in NetCDF-4/HDF5 data for scientific floating-point datasets. They also proposed a Digital Rounding algorithm and compare it with Sz and Bit Grooming algorithm on two synthetic datasets and two real-world datasets.

**Comments:**

- Introduction: as mentioned in this section, FPZIP and ZFP are efficient and effective lossy compressors. Is there any reason why didn't evaluate them? I noticed that referee 1 also asked this same question. Actually, FPZIP is not only lossless but also lossy compression with precision mode. I suggest testing ZFP/FPZIP or briefly explain why FPZIP/ZFP are opted out (e.g., worse than Sz).

Indeed, (Tao et al, 2017a) reported better rate-distortion results for Sz than for FPZIP and ZFP. This is why we chose to evaluate the Sz algorithm rather than the other two. We now clarify this is section 2. We have added the following sentence:

"We chose to evaluate the Sz algorithm because it provides better rate-distortion results than FPZIP and ZFP, see (Tao et al, 2017a)."

- Compression algorithms: the difference of Decimal Rounding and Digit Rounding algorithm seems blurred to me. Can you explain more details about the Decimal Rounding algorithm?

The difference relies in the quantization factor. The Decimal Rounding algorithm uses the same quantization factor to process all the samples. The Digit Rounding algorithm adapts the quantization factor to each sample value. This has been clarified in section 3:

"The difference with the Decimal Rounding algorithm and with Sz's error-controlled quantization is that the Digit Rounding algorithm adapts the quantization factor to each sample value to achieve a specified relative precision of *nsd* digits."

- Application: is there any reason/motivation to compress CFOSAT and SWOT datasets? It's better to explicitly describe the (projected) data volumes that will be produced by these missions.

Thank you for your comment. We have added a part with data volumes for each mission to better described the datasets and better understand the objectives/motivation to compress CFOSAT and SWOT datasets. In section 6.1:

"By the end of the mission in 2023/2024, CFOSAT will have generated about 350TB of data. Moreover, during routine phase, the users should have access to the data less three hours after their acquisition. The I/O and compression performance are thus critical."

And in section 6.2:

"The SWOT mission will generate about 20PB of data during the mission lifetime. Compression algorithms and performance are thus very critical. The mission center is currently defining files format and structure, and thus in this section we evaluated different compression options."

- Conclusion: This statement seems unconvincing to me: "Sz provides higher compression ratios than Decimal Rounding on most datasets. However, for the compression of real scientific datasets, its usability is reduced by the fact that only one error bound can be set for all the variables composing the dataset. It cannot be easily configured to achieve the precision required variable per variable." Actually, Sz provides value-range-based relative error bound, it can apply different absolute error bounds to different variables in the dataset based on the value range of each variable. Please explicitly explain if this mode can satisfy your demand.

The value-range-based relative error bound option (*relBoundRatio*) offered by Sz is different of the absolute error bound (*absErrBound*) and is not applicable to the absolute error-bounded mode. The value-range-based relative error bound limits the decompression errors by considering the global data value range size, whereas the absolute error bound limits the errors to an absolute bound.

An example is provided in Sz User Guide to better understand the value-range-based relative error bound: "Suppose *relBoundRatio* is set to 0.01, and the data set is {100,101,102,103,104,...,110}. In this case, the maximum value is 110 and the minimum is 100. So, the global value range size is 110-100=10, and the error bound will be 10\*0.01=0.1, from the perspective of "relBoundRatio"."

Moreover, two variables in the dataset may have comparable value ranges, but users may require a higher precision for one variable and thus different absolute error bounds.

Nevertheless, the text has been slightly modified:

"However for the compression of netCDF/HDF5 datasets composed of several variables, its usability is reduced by the fact that only one absolute error bound can be set for all the variables. It cannot be easily configured to achieve the precision required variable per variable."

- There are still some grammatical issues that need to be fixed, e.g., "Sz provide" -> "Sz provides", "the compressions obtained" -> "the compression ratios obtained", etc.

Thank you for finding these issues.

They have been corrected.

- The major concern of this paper is the generality and novelty of Digital Rounding algorithm. Based on the experiments, it looks Digital Rounding algorithm is comparatively as good as other approaches and better in some aspects, but the evaluation is only conducted on two datasets/applications. Also, this algorithm looks very similar to Sz's linearly scaling quantization. If not, the paper should explicitly explain the difference.

The error-controlled quantization described in (Tao et al, 2017a) is a linear quantization of the Sz's "first-phase prediction" error. The similarity with the Digit Rounding algorithm is that it the quantization error is bounded. The main difference is that in the case of the Sz's error-controlled quantization, the error bound is a fixed value (both in the absolute error bounded mode and in the value-range-based relative error bounded mode) whereas in the Digit Rounding algorithm, the quantization factor depends on each sample value to achieve the specified relative precision of *nsd* digits. The following explanation has been added in the text:

"The Digit Rounding algorithm [...] is also similar to Sz's error-controlled quantization (Tao et al, 2017a) in the sense that the quantization error is bounded. The difference with the Decimal Rounding

algorithm and with Sz's error-controlled quantization is that the Digit Rounding algorithm adapts the quantization factor to each sample value to achieve a specified relative precision of *nsd* digits."

**Reply to Anonymous Referee #1 (Report #2)**

We are grateful to the referee for her/his constructive and thorough criticism and suggestions to our manuscript.

The authors clearly put a lot of effort into this revision, and the quality of the manuscript has greatly improved. I am overall satisfied with their responses to my comments and suggestions. In particular, the additional details on the algorithms and test data are much appreciated as is the improvement in focus. The contributions have also been clarified. Finally, I'll note that the writing quality is very much improved for this revision. For subsequent manuscripts, I would encourage the authors to aim for this higher quality of writing in the initial submission.

A few minor issues:

p.3, line 13: "Both algorithms..." - which two are being referred to? Or is it to all three (in which case, it should by "All three algorithms...")

This has been replaced by:

"LZ77 and Huffman coding exploit different types of redundancies ..."

*p.3, line 29: "... to degrade the least significant ..." This use of "degrade" is a bit awkward - consider rephrasing - maybe "to remove the least significant"*

It has been reworded as you have suggested:

"The Bit Grooming algorithm creates a bitmask to remove the least significant bits of the mantissa of IEEE 754 floating-point data."

p.4, line 29: "consists in" => "consists of"

It has been corrected.

*p.* 5: Section 3 (line 16) refers to Table 3. Note that Table 3 lists mean error, mean absolute error, and maximum absolute error. However, these metrics are not defined until Section 4. Should section 3 and 4 be switched? Or a forward reference added?

A forward reference has been added to section 4 and in the caption of Table 3:

"Table 3 provides the maximum absolute errors, the mean absolute errors and the mean errors (defined in section 4) obtained with varying nsd values ..."

"Table 3: Maximum absolute errors, mean absolute errors and mean errors of the Digit Rounding algorithm preserving a varying number of significant digits (nsd) on an artificial dataset composed of 1,000,000 values evenly spaced over the interval [1.0, 2.0). The error metrics are defined in section 4."

p.5: Section 3 also refers to Table 4, which uses CR -- which is not defined until Section 4. Though it seems that in Table 4, CR is a percentage as opposed to a ratio as defined in Section 4, so this needs to be clarified.

Clarification has been added in the caption of Table 4:

"Table 4: Comparison of the Bit Grooming and Digit Rounding algorithms for the compression of a MERRA dataset. Shuffle and Deflate (level 1) are applied. The compression ratio (CR) is defined by the ratio of the compressed file size over the reference data size (244.3MB) obtained with Deflate (level 5) compression. Bit Grooming results are extracted from (Zender, 2016a)."

p.5, line 19: "...a relative error ..." I think you mean "absolute error"

You are right. We mean an absolute error here.

It has been corrected.

p.10, line 10: "compressions" => "compression ratios"

p. 13, line 27: "was configured: => "were configured"

p.13, line 34: "provides" => "provide"

p. 14, line 23: "fails achieving" => "fails to achieve"

Thank you for finding these issues.

They have all been corrected.

**Reply to Charlie Zender**

We are grateful to the referee for her/his constructive and thorough criticism and suggestions to our manuscript.

*First, thank you for addressing my concerns. I have never seen more changes made to a manuscript in review! The manuscript is significantly improved and more readable.*

I have two minor suggestions I hope you will incorporate:

1. NetCDF should only be capitalized when it begins a sentence. This affects the manuscript title and dozens of instances in the body. The authoritative spelling guide to netCDF is at the bottom of this: https://www.unidata.ucar.edu/software/netcdf/docs/BestPractices.html

Thank you for pointing this spelling out. It has been corrected all over the manuscript.

2. The NCO/ncks commands shown in the supplementary materials could be significantly shortened by using regular expressions for the variable names, e.g.,

ncks --ppc incidence\_.?=.5

instead of

ncks --ppc incidence\_1=.5 --ppc incidence\_2=.5 --ppc incidence\_3=.5 --ppc incidence\_4=.5 --ppc incidence\_5=.5

Moreover, one can set a default precision and only give per-variable exceptions to that precision. Also most people shorten --overwrite to -O because it is used so frequently.

The option --overwrite has been replaced by -O in all the calls to ncks that are given in the supplement. The calls to ncks for the compression of COSAT and SWOT datasets given in the supplement have been shorten using the default precision option "--pcc default=" and ".?" for regular expressions in the variable names:

```
ncks -O -4 -L 1 --ppc default=.3 --ppc cycle duration=.7 --ppc echo 11 0.?=.12 \
--ppc elevation .?=.5 --ppc incidence .?=.5 --ppc ly=.2 --ppc mispointing=.14 \
--ppc pri=.1 --ppc pseudo misp=.14 \
TMP_TEST_SWI_L1A ____F_20160830T150000_20160830T164500_dr.nc)
ncks -4 -L 1 -O SWOT L2 HR PIXC decomp.nc SWOT L2 HR PIXC bg.nc --ppc default=8 \
--ppc azimuth index=4 --ppc classification=3 --ppc cross track=15 \
--ppc dheight .?=6 --ppc dlook dphase .?=6 --ppc dphase=6 \
--ppc dry tropo range correction=4 --ppc height=6 --ppc ifgram imag=15 \
--ppc ifgram_real=15 --ppc illumination_time=15 --ppc instrument_.?=6 \
--ppc ionosphere_range_correction=6 --ppc latitude=15 --ppc longitude=15 \
--ppc num rare looks=2 --ppc phase screen=6 --ppc pixel area=11 --ppc range index=4
--ppc sensor s=11 --ppc solid earth tide height correction=4 \
--ppc wet tropo range correction=4 --ppc xover roll correction=4
ncks -4 -L 1 -0 pixel cloud decomp.nc -o pixel cloud bg.nc --ppc default=8 \
--ppc azimuth index=4 --ppc classification=3 --ppc cross track=15 \
--ppc dlatitude dphase=5 --ppc dlongitude dphase=5 --ppc height=6 --ppc ifgram=7 \
--ppc illumination time=15 --ppc latitude=15 --ppc longitude=15 \
--ppc num med looks=3 --ppc num_rare_looks=2 --ppc phase_noise_std=7 \
--ppc pixel_area=11 --ppc power_left=7 --ppc range_index=4 --ppc regions=7 \
--ppc sigma0=7 --ppc x factor.?=7
```

**Evaluation of lossless and lossy algorithms for the compression of scientific datasets in NetCDFnetCDF-4 or HDF5 files**

Xavier Delaunay1, Aurélie Courtois1, Flavien Gouillon2

1Thales, 290 allée du Lac, 31670 Labège, France

2CNES, Centre Spatial de Toulouse, 18 avenue Edouard Belin, 31401 Toulouse, France

Correspondence to: Xavier Delaunay (xavier.delaunay@thalesgroup.com)

Abstract. The increasing volume of scientific datasets requires the use of compression to reduce data storage and transmission costs, especially for the oceanographic or meteorological datasets generated by Earth observation mission ground segments. These data are mostly produced in NetCDFnetCDF files. Indeed, the NetCDFnetCDF-4/HDF5 file formats are widely used throughout the global scientific community because of the useful features they offer. HDF5 in particular offers a dynamically loaded filter plugin so that users can write compression/decompression filters, for example, and process the data before reading or writing them to disk. This study evaluates lossy and lossless compression/decompression methods through NetCDFnetCDF-4 and HDF5 tools on analytical and real scientific floating-point datasets. We also introduce the Digit Rounding algorithm, a new relative error-bounded data reduction method inspired by the Bit Grooming algorithm. The

15 Digit Rounding algorithm offers a high compression ratio while keeping a given number of significant digits in the dataset. It achieves a higher compression ratio than the Bit Grooming algorithm with slightly lower compression speed.

**1** Introduction**

20

5

Ground segments processing scientific mission data are facing challenges due to the ever-increasing resolution of on-board instruments and the volume of data to be processed, stored and transmitted. This is the case for oceanographic and meteorological missions, for instance. Earth observation mission ground segments produce very large files mostly in <a href="https://www.netCobe.com">NetCobe.netCobe</a> format, which is standard in the oceanography field and widely used by the meteorological community. This file format is widely used throughout the global scientific community because of its useful features. The fourth version of the <a href="https://www.netCobe.com">NetCobe.netCobe.netCobe.netCobe</a> library, denoted <a href="https://www.netCobe.com">NetCobe.netCobe</a> library, denoted <a href="https://www.netcobe.com">NetCobe.netCobe</a> library, denoted <a href="https://www.netcobe.com">NetCobe.netCobe</a> library, denoted <a href="https://www.netcobe.com">https://www.netcobe.com</a> segments produce very large files mostly in <a href="https://www.netcobe.com">NetCobe.netCobe</a> format, which is standard in the oceanography field and widely used by the meteorological community. This file format is widely used throughout the global scientific community because of its useful features. The fourth version of the <a href="https://www.netcobe.com">NetCobe.netCobe.netCobe.com</a> library, denoted <a href="https://www.netcobe.com">NetCobe.netCobe.netCobe.com</a> library, denoted <a href="https://www.netcobe.com">https://www.netcobe.com</a> library 
[revised manuscript text omitted]

| Sign | Exponent | Mantissa                                | Decimal     | Notes                |
|------|----------|-----------------------------------------|-------------|----------------------|
| 0    | 1000000  | 10010010000111111011011                 | 3.14159265  | Exact value of $\pi$ |
| 0    | 1000000  | 10010010000111111011011                 | 3.14159265  | nsd = 8              |
| 0    | 1000000  | 10010010000111111011010                 | 3.14159250  | nsd = 7              |
| 0    | 1000000  | 10010010000111111010000                 | 3. 14159012 | nsd = 6              |
| 0    | 1000000  | 10010010000111110000000                 | 3. 14157104 | nsd = 5              |
| 0    | 1000000  | 10010010000100000000000                 | 3.14111328  | nsd = 4              |
| 0    | 1000000  | 100100101000000000000000000000000000000 | 3. 14453125 | nsd = 3              |
| 0    | 1000000  | 100101000000000000000000000000000000000 | 3.15625000  | nsd = 2              |
| 0    | 1000000  | 110000000000000000000000000000000000000 | 3. 50000000 | nsd = 1              |
| 0    | 1000000  | 000000000000000000000000000000000000000 | 4.00000000  | nsd = 0              |

Table 3: Maximum absolute errorsandmean absolute errorsandmean errorsof the Digit Rounding algorithm preserving a10varying number of significant digits (*nsd*) on an artificial dataset composed of 1,000,000 values evenly spaced over the interval [1.0,2.0). The error metrics are defined in section 4.

[revised manuscript text omitted]

**5 Figure 2: The Digit Rounding algorithm.**